# Influences on Chemical Distribution Patterns across the west Greenland Shelf: The Roles of Ocean Currents, Sea Ice Melt, and Freshwater Runoff

Claudia Elena Schmidt[1,2], Tristan Zimmermann[3], Katarzyna Koziorowska[4], Daniel Pröfrock[3], Helmuth Thomas[1,2]

[1]Helmholtz-Zentrum Hereon, Institute of Carbon Cycles, 21502, Geesthacht, Germany
[2]Carl von Ossietzky University Oldenburg, Institute for Chemistry and Biology of the Marine Environment, 26129, Oldenburg, Germany
[3]Helmholtz-Zentrum Hereon, Institute of Coastal Environmental Chemistry, 21502, Geesthacht, Germany
[4]Institute of Oceanology Polish Academy of Sciences, Department of Marine Chemistry and Biochemistry, 81-712, Sopot, Poland

*Correspondence to*: Helmuth Thomas (helmuth.thomas@hereon.de)

**Abstract** The west Greenland shelf is a dynamic marine environment influenced by various physicochemical and biological processes. This study provides a comprehensive overview of the main factors affecting the distribution of macronutrients, carbonate system parameters, and dissolved trace elements during July. Key drivers include major ocean currents, melting sea ice, and terrestrial freshwater runoff, each contributing uniquely to the cycling and spatial distribution of chemical constituents. Major ocean currents, such as the southward-moving Baffin Island Current (BIC) and the northward-moving West Greenland Current (WGC), introduce water masses with distinct chemical signatures that shape the chemical composition of shelf waters. Melting sea ice serves as an important source of freshwater and dissolved constituents for the marine environment. During the study period, we were able to capture a distinct nutrient gradient following the east-to-west direction of sea ice retreat, with low nutrient levels in highly productive shelf waters and high nutrient levels in areas with prolonged ice cover. This process also influenced the carbonate system, leading to changes in pH and aragonite saturation states, which is critical to the health of marine organisms. Terrestrial freshwater runoff, particularly from the Greenland Ice Sheet (GIS), replenishes macronutrients in the photic zone, stimulating primary production and creating important $CO_2$ sinks. However, coastal surface waters become more susceptible to acidification by the input of poorly buffered glacial freshwater. Understanding these key drivers is essential for predicting future changes in the marine chemistry and ecosystem dynamics on the west Greenland shelf, especially in the context of ongoing climate change within this high-latitude region.

# 1 Introduction

The physical system of the Arctic Ocean is inevitably changing, with profound implications for primary productivity (PP) and the biogeochemical cycling of nutrients and carbon across continental shelves and deep basins (Juranek, 2022). The complex coastal current system that connects the continental shelves of Greenland with the Arctic Ocean, the North Atlantic Ocean, and the Pacific Ocean strongly impacts PP (Vernet et al., 2021), creating an important environment for biota and fisheries (Krawczyk et al., 2021). The major ocean currents on the west side of Greenland are shown in Fig. 1 a. The circulation across

the west Greenland shelf and Baffin Bay is dominated by two major currents in the region: the southward-moving Baffin Island Current (BIC) along the Baffin Island continental slope and the northward-moving West Greenland Current (WGC) along the west Greenland continental slope (Rysgaard et al., 2020). On the western side of Baffin Bay, the surface-intensified BIC transports water from the Alaskan shelf and the western Beaufort Sea through the narrow and shallow channels of the Canadian Arctic Archipelago (CAA), including Nares Strait, Jones Sound, and Lancaster Sound, across Davis Strait into the Labrador

Sea (Tang et al., 2004; Curry et al., 2011; Aksenov et al., 2016). Arctic waters of Pacific origin that enter Baffin Bay through the CAA have been identified as sources of dissolved trace elements (e.g., dMn, dFe, dCu, dNi) (Colombo et al., 2020; Colombo et al., 2021; Jensen et al., 2022) as well as carbon (Azetsu-Scott et al., 2010; Shadwick et al., 2011b; Burgers et al., 2024). Further south, Davis Strait acts as a gateway for the export of dissolved trace elements (Krisch et al., 2022a) and dissolved inorganic nitrogen (Juranek, 2022) from the Arctic to the North Atlantic Ocean. On the eastern side of Baffin Bay,

the West Greenland Current (WGC) flows northward across Davis Strait, consisting of the warm, low-salinity West Greenland Coastal Current (WGCC) on the shelf and the warm, high-salinity West Greenland Slope Current (WGSC) (Curry et al., 2011). These currents transport waters of North Atlantic origin, which have been associated with anthropogenically elevated concentrations of dissolved lead (dPb) (Colombo et al., 2019), as well as enhanced biological productivity and species richness across the west Greenland shelf (Krawczyk et al., 2021).

In the last two decades, sea ice concentrations in Baffin Bay and the Labrador Sea have declined significantly due to climate change (Krawczyk et al., 2021), resulting in an earlier melt onset and delayed freeze-up (Stroeve and Notz, 2018; Ballinger et al., 2022). The reduction in sea ice cover, combined with increased inflow of Atlantic-sourced waters (Krawczyk et al., 2021) and elevated freshwater discharge (Møller et al., 2023), has been associated with an overall increase in PP and species richness along the west Greenland shelf (Krawczyk et al., 2021; Møller et al., 2023). In all seasonally ice-covered areas, intense but

short-lived phytoplankton blooms develop in low-salinity waters at the edge of melting and retreating sea ice from spring to late summer (Niebauer et al., 1995; Perrette et al., 2011). Phytoplankton growth is triggered by the stabilizing effect of sea ice meltwater-induced stratification and the increased solar irradiance as the ice cover shrinks (Strass and Nöthig, 1996; Perrette et al., 2011). The newly exposed waters are initially nutrient-rich but become rapidly depleted following the onset of the bloom (Niebauer et al., 1995). In early summer, macronutrient distributions (NOx = nitrate + nitrite; phosphate, $PO_4^{3-}$; silicate,

$Si(OH)_4$) in surface waters of Baffin Bay decrease along a west-to-east gradient, generally following the sea ice coverage (Tremblay et al., 2002; Lafond et al., 2019). Although sea ice meltwater is considered a negligible net source of macronutrients,

it can significantly enhance trace element concentrations in the receiving waters (Hölemann et al., 1999; Tovar-Sánchez et al., 2010; Kanna et al., 2014; Evans and Nishioka, 2019). Elevated trace element concentrations in Arctic sea ice have been linked to entrained sediments (Measures, 1999) or atmospheric deposition from continuous emissions in highly industrialized countries (McConnell and Edwards, 2008). Sea ice meltwater also influences the carbonate system of Baffin Bay by increasing alkalinity (AT) relative to salinity. When the effect of salinity is removed, sea ice meltwater has been shown to supply additional AT in the form of buffering ions such as carbonate ($CO_3^{2-}$) (Jones et al., 1983; Fransson et al., 2023). During sea ice formation, ikaite ($CaCO_3 \cdot 6H_2O$) precipitates within the sea ice and is reintroduced to the water column during ice melt. This dissolution process results in higher AT concentrations relative to dissolved inorganic carbon (CT), lower partial pressure of $CO_2$ ($pCO_2$), and higher pH and aragonite saturation state ($\Omega$ Aragonite) in surface waters (Rysgaard et al., 2011; Rysgaard et al., 2012; Fransson et al., 2013). However, on an annual cycle, sea ice does not serve as a net source or sink of these species to the underlying seawater. Instead, it serves as a mechanism for temporal and spatial redistribution, as these species are trapped during ice formation in autumn and winter, and subsequently released during ice melt – potentially at a different location due to drifting sea ice (Thomas et al., 2011).

The mass loss of the Greenland Ice Sheet (GIS) has increased sixfold since the 1980s, with the region north of Davis Strait experiencing the largest mass loss (Mouginot et al., 2019; Wouters and Sasgen, 2022). The export of glacial runoff from the GIS modifies the delivery of solutes and nutrients to near coastal regions (Hawkings et al., 2015) and the continental shelf (Cape et al., 2019). This glacier-derived freshwater alters marine concentrations of macronutrients (Hawkings et al., 2015; Hawkings et al., 2016; Hawkings et al., 2017; Hendry et al., 2019) and dissolved trace elements (Bhatia et al., 2013; Hawkings et al., 2020; Krause et al., 2021), influences local hydrography, and promotes the upwelling of nutrient-rich deep waters (Juul-Pedersen et al., 2015; Meire et al., 2017). These processes enhance marine productivity and extend the duration of the annual productive season (Oksman et al., 2022). However, glacial freshwater is characterized by low AT and carbonate ion concentration relative to marine waters, thereby reducing the capacity of glacially modified waters to resist changes in pH driven by atmospheric $CO_2$ uptake (Bates and Mathis, 2009; Chierici and Fransson, 2009; Fransson et al., 2015). Along the Greenlandic coast, AT dilution from glacial meltwater has been observed to drive corrosive conditions in surface waters, resulting in decreased $\Omega$ Aragonite and pH (Shadwick et al., 2013; Henson et al., 2023). Photosynthetic $CO_2$ uptake can partially mitigate this negative effect of AT dilution, as PP reduces CT and $pCO2$ concentrations in surface waters and increases the carbonate saturation state (Chierici and Fransson, 2009; Meire et al., 2015; Hopwood et al., 2020).

In this study, we present recent measurements (July 2021) of the marine carbonate system, nutrient concentrations and dissolved (d; < 0.45 μm) trace element concentrations across the west Greenland shelf in southern Baffin Bay between 64°N and 71°N (Fig. 1 b). Given the above considerations, the objectives of this study are to: (a) understand the physicochemical processes that influence the internal cycling of chemical constituents within the distinct water masses of the shelf; and (b) resolve the regional and spatial differences in the distribution patterns on the west Greenland shelf during late summer that are driven by (i) major ocean currents, (ii) melting sea ice, and (iii) terrestrial freshwater runoff.

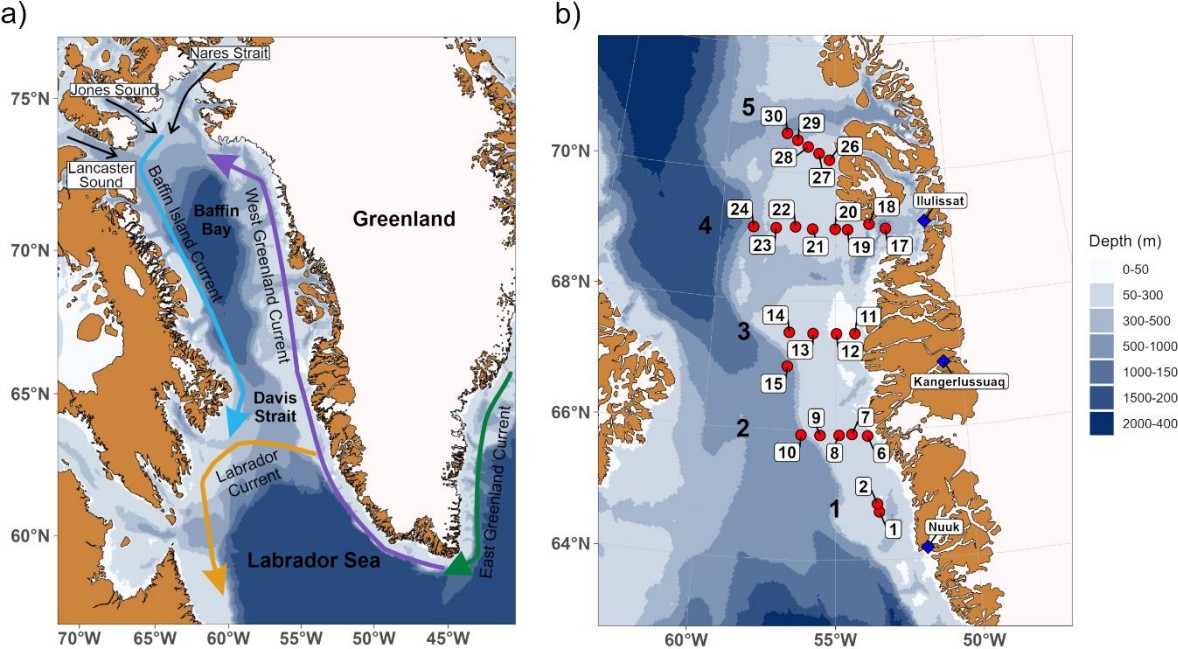

**Figure 1: a) Schematic overview of the ocean currents and bathymetry in Baffin Bay and Davis Strait. Colored lines with arrows indicate the direction of different ocean currents. Bathymetry data from the GEBCO 2023 grid (GEBCO Bathymetric Compilation Group 2023). b) Map of the study area during DANA 6/21 with red points indicating the sampling locations and their affiliation to a certain transect (1-5).**

## 2 Material and Methods

### 2.1 Study Area

A map of the study area with stations indicated by red dots is given in Fig. 1 b. General information about the selected stations and samples used in this study is provided in Table S 1. Samples were taken along two transects to the south and three transects to the north of Davis Strait along the west Greenland shelf. Transect 1 was located in a shallow area (< 120 m depth) north of Nuuk. Transects 2 and 3 started close to the mouth of Kangerlussuaq Fjord and Nassuttooq Fjord, respectively. Station 15, which is part of transect 3, had to be moved further south because of the presence of sea ice. Transect 4 started in Disko Bay approximately 70 km from the mouth of the Ilulissat Icefjord, where the Jakobshavn Isbræ terminates. Transects 2, 3, and 4 continued westward away from the coast towards the shelf edge. Transect 5 started close to Disko Island and continued towards the northwest.

### 2.2 Sampling

Water column samples were taken at 25 stations as conductivity-temperature-depth (CTD) casts along five transects during ten consecutive sampling days between 18 and 28 July 2021 on board RV *Dana* (DANA 6/21, Fig. 1 b, Table S 1). A SBE 911 CTD+ unit (Sea-Bird Scientific; Bellevue, USA) was used to collect hydrographic profiles, including temperature, salinity,

pressure, and dissolved oxygen. Two oxygen sensors were part of the CTD system and calibrated prior to the cruise. By cross-referencing the data of the two sensors, we were able to monitor their drift during the cruise and correct the data accordingly. Samples were collected in 10 L metal-free Niskin water samplers (Ocean Test Equipment; Fort Lauderdale, USA). Detailed information about each sample is given in Table S 1, with sample IDs corresponding to the respective station and sampling depth.

### 2.2.1 Macronutrients

Water samples for nutrient analysis were collected in acid-washed 100 mL HDPE (high-density polyethylene) bottles. The bottles were preconditioned three times with sample water before being filled. The samples were stored frozen at -20°C without filtration until analysis.

### 2.2.2 Dissolved trace elements

Bulk samples for multi-element analysis were filled into acid-cleaned 2 L HDPE bottles. The bottles were rinsed three times with sample water before being filled. Immediately after sampling, the water samples were filtered as duplicates using high-purity, metal-free PFA (Perfluoroalkoxy alkane) filtration bombs (Savillex, Eden Prairie, USA) pressurized with nitrogen 5.0 (Air Liquide Danmark; Taastrup, Denmark) and acid-washed polycarbonate filters (0.45 μm, 47 mm diameter, Whatman, UK) in a clean bench (Erlab; Val de Reuil Cedex, France). More information about the filtration procedure can be found in Przibilla *et al.* (2023). The filtered samples were collected in pre-cleaned 250 mL HDPE bottles and stabilized using trace metal grade HCl (Fisher Scientific; Waltham, USA) before storage and shipping.

### 2.2.3 Carbonate system parameters

Samples for AT and CT were collected in 300 mL BOD (biological oxygen demand) bottles (Environmental Express, Charleston, USA) with an addition of 300 μL saturated mercury chloride solution (Carl Roth, Karlsruhe, Germany). The bottles were sealed with ground-glass stoppers (Wheaton Science Products, New Jersey, USA), silicone- and halogen-free grease (type M, Apiezon, Manchester, UK), and plastic caps (Wheaton Science Products, New Jersey, USA), leaving no headspace. The samples were stored in darkness at ambient temperature until analysis.

## 2.3 Sample preparation and chemical analysis

### 2.3.1 Macronutrients

Nitrite, phosphate, and silicate were measured on a SmartChem 200 discrete analyser (AMS Alliance, Rome, Italy) following the reagent concentrations for the manual method in Hansen and Koroleff (1999), with the inclusion of a 150 g L$^{-1}$ solution of sodium dodecylsulfate as a dispersant for phosphate analysis. Nitrate was measured following the vanadium chloride reduction

technique described by Schnetger and Lehners (2014). Community reference material for nutrients in seawater supplied by Kanso Technos Co. LTD, Japan, was used for quality control.

### 2.3.2 Dissolved trace elements

All equipment for trace metal analysis was acid-washed prior to use and rinsed with type I reagent-grade water (resistivity: 18.2 MΩ·cm) obtained from a Milli-Q Integral water purification system (Merck; Darmstadt, Germany) to prevent contamination. Dissolved trace elements (dV, dFe, dMn, dCo, dNi, dCu, dCd, and dPb) in seawater were measured with a seaFAST SP2 system (Elemental Scientific; Omaha, USA) coupled online to an inductively coupled plasma mass spectrometer (ICP-MS) (Agilent 7900, Agilent Technologies; Tokyo, Japan). The seaFAST SP2 system contained two columns filled with
Nobias chelate-PA1 resin (HITACHI High-Tech Fielding Corporation; Tokyo, Japan). Detailed operating parameters and instrument configurations are given in Table S 2. More information about the analytical procedure can be found in Ebeling *et al.* (2022). For method validation, the certified reference material (CRM) NASS 7 (National Research Council Canada; Ottawa, Canada) as well as an in-house reference material (KBA-QC) mixed from single element standards (Carl Roth GmbH, Karlsruhe, Germany or Sigma-Aldrich, Missouri, USA) and custom-made multi-element standards (all traceable to NIST
standards) of different compositions (Inorganic Ventures, Christiansburg, USA) were used. Recovery rates (between 97 % and 119 %) are given in Table S 3.

### 2.3.3 Carbonate system parameters

Measurements of AT and CT were performed using a VINDTA 3 C system (Marianda; Kiel, Germany). AT and CT were determined simultaneously by potentiometric titration using an 800 Dosino (Metrohm; Filderstadt, Germany) with an
Aquatrode plus (Metrohm; Filderstadt, Germany) and coulometric titration using a CM5017O coulometer (UIC; Illinois, USA), respectively. Both instruments were calibrated against seawater CRMs (Scripps Institution of Oceanography; San Diego, USA) to ensure a precision of $\pm$ 1 µmol kg$^{-1}$ for each parameter.

### 2.4 Data analysis

The raw CTD data were processed and averaged into 0.5 dbar pressure bins. The processed CTD data were used to construct
water depth profiles of potential temperature, salinity, oxygen (Fig. 2), and potential density (Fig. S 1). Calculations were performed using R (version 4.2.2; (R Core Team, 2022)) and RStudio (version 2023.06.0; (Posit team, 2023)). Bivariate, linear data interpolation in combination with a surface approximation using multilevel B-splines with the "interp" function (package akima; (Akima and Gebhardt, 2022)) and the "mba.surf" function (package MBA; (Finley et al., 2022)) was used before plotting.

For trace elements analysis, filtration duplicates were obtained and measured as triplicates ($n = 6$). Outliers were identified using a Dean-Dixon outliers test and removed from further analysis. Mean concentrations ($\mu$) and standard deviation (*SD; $\sigma$*) were estimated, considering twice the *SD* to represent $\mu \pm 2\sigma$. Limits of detection (LOD) and limits of quantification (LOQ)

are given in Table S 3 and were calculated according to DIN 32645:2008-11, based on measurements of filtrations blanks ($n = 8$). The LOD is defined as three times the standard deviation ($3 \times$ SD) of the blank measurements and represents the lowest concentration that can be reliably distinguished from background noise, though not necessarily quantified with precision. The LOQ is defined as ten times the standard deviation ($10 \times$ SD) and represents the lowest concentration that can be quantitatively determined with acceptable accuracy and precision (DIN e.V., 2008).

The sum parameter NOx for nitrate and nitrite (NOx = nitrate + nitrite) was established and used throughout the discussion. The LOQs for the nutrient analysis are given in Table S 3.

Parameters of the carbonate system (pH, $p$CO$_2$, Revelle factor, and $\Omega$ Aragonite) were calculated using the CO2Sys Macro (Pierrot et al., 2011) with salinity, temperature, AT and CT as input variables. As input parameters, we used the dissociation constant by Mehrbach et al. (1973), refit by Dickson and Millero (1987), the HSO$_4^-$ dissociation constant by Dickson (1990) and the seawater pH-scale (Dickson, 1990; Dickson and Millero, 1987; Mehrbach et al., 1973). We used the CO2Sys Macro to calculate temperature-normalized CT (CT$_{temp}$) according to Wu *et al.* (2019). The calculation was based on the median potential temperature of WGSW (3.23°C, $n = 465$) from the CTD profiles, filtered for a depth range of 30 to 50 m. Further input parameters were observed salinity, AT, and the in situ $p$CO$_2$.

For statistical analysis, the data of the different parameters were merged according to station and sampling depth, obtaining one data matrix with 17 parameters and 41 observations. As a minor proportion of the total observations (< 3 %) were below the LOD or LOQ, a single imputation method was applied (Helsel, 2005; Jain, 2016). Values below the LOD were replaced with a random value between zero and the LOD, while values below the LOQ were replaced with a random value between the LOD and LOQ. Each parameter was tested for normality (Shapiro-Wilk test, $\alpha = 0.01$) and transformed (log- or Box Cox-transformation) if the normality criteria was not met (Table S 4). The data were standardized, and principal component analysis (PCA) with varimax rotation was performed using the "principal" function (package psych; (Revelle, 2024)). Three principal components (PCs) were selected based on the Guttman-Kaiser criterion and the scree plot, retaining only those with eigenvalues greater than 1 (Bro and Smilde, 2014). A broken stick analysis was performed to evaluate the significance of variable loadings and identify which variables are associated with specific PCs (Peres-Neto et al., 2003).

## 3 Results

The distributions of basic oceanographic parameters (salinity, potential temperature, oxygen, apparent oxygen utilization (AOU)), macronutrients (NOx, silicate, phosphate), dissolved trace elements (dV, dMn, dFe, dCo, dNi, dCu, dCd, and dPb), and carbonate system parameters (AT, CT, pH, $p$CO$_2$, Revelle factor, $\Omega$ Aragonite) are presented in Sect. 3.1 to 3.4. In Fig. 2, detailed water column profiles of salinity, potential temperature, and oxygen are provided. An overview of minimum and maximum values of macronutrients, carbonate system parameters, and trace elements are presented in Table 1. Figures 3 to 5 illustrate the vertical distributions of macronutrients, dissolved trace elements, and carbonate system parameters for each station and transect. Surface water concentration plots for individual parameters are provided in the supplementary material

(Fig. S 2 to Fig. S 4). In Sect. 3.5, we classify the water masses present across the west Greenland shelf based on characteristic salinity and potential temperature ranges.

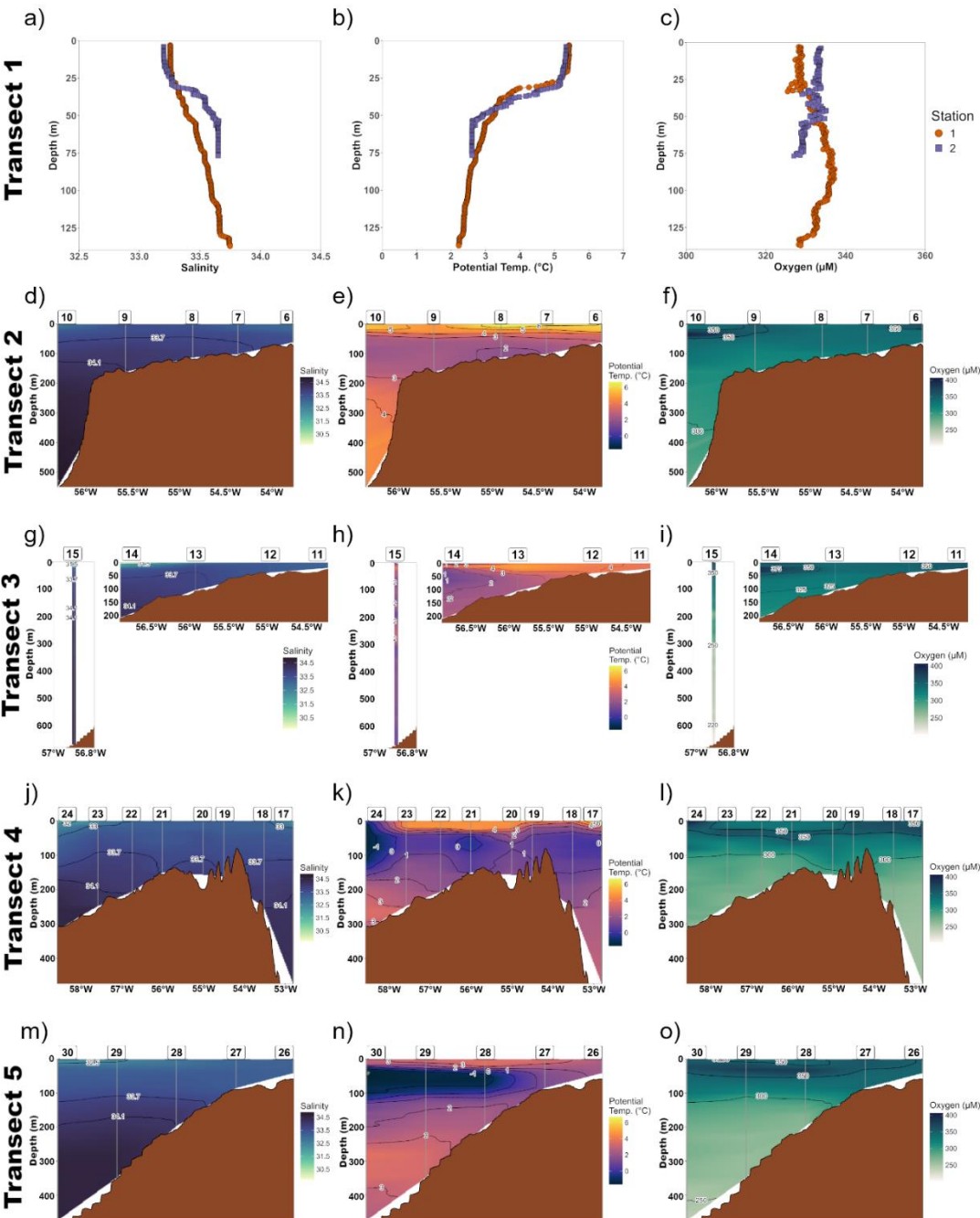

**Figure 2: Water depth profiles illustrating the spatial distribution of salinity, potential temperature, and oxygen along transect 1 (a-c), transect 2 (d-f), transect 3 (g-i), transect 4 (j-l), and transect 5 (m-o). For transect 1, data are not shown as interpolated profiles**
**due to limited data availability. For transect 3, station 15 is displayed separately because of its southerly location (refer to Fig. 1 b).**

## 3.1 Basic oceanographic parameters

The distribution of salinity, potential temperature, and oxygen are shown as water depth profiles for each transect in Fig. 2. Ranges of minimum and maximum values for salinity, potential temperature, oxygen and AOU are summarized in Table S 5. Salinity was generally low in surface waters at stations close to the coast (stations 6, 11 and 17) and towards the west of each transect (stations 14, 15, 24, 29 and 30), with a minimum value of 29.70 recorded at the surface of station 14 (Fig. 2 g). The highest salinity (34.89) was observed in deep waters at station 10, between 530 and 558 m depth (Fig. 2 d). Potential temperature was highest in surface waters in the southeastern part of Davis Strait, particularly at transect 2 (stations 6, 7, 8, and 10), and at station 17 in Disko Bay. Lower potential temperatures were found further offshore in the northwestern part of Davis strait. The minimum potential temperature of -1.63 °C was recorded in intermediate waters (40 to 55 m) at station 30 (Fig. 2 n), while the maximum value of 6.7 °C occurred in surface waters at station 8 (Fig. 2 e). In general, higher oxygen concentrations and lower AOU values were found in the photic zone (1.5 to 40 m), either near the Greenlandic coast at the beginning of transects (stations 6 and 17) or at the western ends (stations 10, 14, 15, and 24). The lowest oxygen concentration (204 $\mu$mol L$^{-1}$) and highest AOU (147 $\mu$mol L$^{-1}$) were observed in deep waters at station 15 (645 to 673 m) (Fig. 2 i). Conversely, the highest oxygen concentration (406 $\mu$mol L$^{-1}$) and lowest AOU (-72 $\mu$mol L$^{-1}$) were measured in intermediate waters (34 to 37 m) at station 10 (Fig. 2 f).

**Table 1: Minimum and maximum values of macronutrients, carbonate system parameters and trace elements across the study area.**

| Parameter | Unit | Minimum | | Maximum | | Literature data |
|---|---|---|---|---|---|---|
| | | Value | Station [Depth in m] | Value | Station [Depth in m] | Min – Max range |
| NOx | µmol L$^{-1}$ | 0.21 | 13 [30] | 17.85 | 15 [667] | 2015: 0.032 – 24.3 [1] <br> 2016: 0.05 – 28.5 [2] |
| Silicate | µmol L$^{-1}$ | 0.51 | 22 [3] | 35.95 | 15 [667] | 2015: 0.32 – 103.7 [1] <br> 2016: 0.11 – 131.4 [2] |
| Phosphate | µmol L$^{-1}$ | 0.12 | 22 [3] | 1.43 | 15 [667] | 2015: 0.46 – 1.97 [1] <br> 2016: 0.06 – 2.2 [2] |
| AT | µmol kg$^{-1}$ | 2062 | 15 [3] | 2305 | 10 [551] | 2015: 2156 – 2294 [4] <br> 2016: 2176 – 2310 [3] |
| CT | µmol kg$^{-1}$ | 1949 | 15 [3] | 2214 | 15 [667] | 2015: 2008 – 2255 [4] <br> 2016: 1937 – 2277 [3] |
| CT:AT | | 0.907 | 17 [3] | 0.968 | 15 [667] | NA |
| pH | | 7.86 | 15 [667] | 8.20 | 15 [32] | 2015: 7.40 – 7.81 [1] |
| $p$CO$_2$ | µatm | 249 | 15 [32] | 559 | 15 [667] | NA |
| Revelle factor | | 11.85 | 17 [3] | 17.80 | 15 [667] | NA |
| $\Omega$ Aragonite | | 0.90 | 15 [667] | 2.17 | 17 [3] | NA |
| dV | ng L$^{-1}$ | 1270 ± 30 | 24 [3] | 1810 ± 60 | 10 [551] | NA |
| dMn | ng L$^{-1}$ | 54.8 ± 1.8 | 24 [317] | 503 ± 20 | 12 [37] | 13.5 ± 1.7 – 306.0 ± 2.6 [1] |
| dFe | ng L$^{-1}$ | 56.9 ± 2.5 | 27 [15] | 630 ± 30 | 18 [256] | 20.0 ± 0.6 – 112 ± 8 [1] |
| dCo | ng L$^{-1}$ | 3.81 ± 0.15 | 10 [551] | 14.7 ± 0.6 | 11 [3] | NA |
| dNi | ng L$^{-1}$ | 232 ± 7 | 6 [3] | 381 ± 4 | 24 [35] | 240 ± 12 – 376 ± 5 [1] |
| dCu | ng L$^{-1}$ | 85.4 ± 2.5 | 10 [551] | 206 ± 3 | 24 [3] | 115 ± 10 – 333.0 ± 0.7 [1] |
| dCd | ng L$^{-1}$ | 3.1 ± 0.7 | 8 [4] | 52.2 ± 1.2 | 15 [667] | 20.0 ± 0.5 – 49.0 ± 0.6 [1] |
| dPb | ng L$^{-1}$ | 0.99 ± 0.08 | 28 [186] | 4.04 ± 0.13 | 10 [551] | 0.52 – 6.0 [1] |

*Note.* The measurement uncertainty of trace elements corresponds to 2 SD. Literature data from Geotraces cruise GN02 in 2015 (station BB1, BB2 and BB3) and GreenEdge cruise in 2016 (stations G100 to G3000). [1](GEOTRACES Intermediate Data Product Group, 2023). [2](Bruyant et al., 2022). [3](Miller et al., 2020). [4](H. Thomas, personal communication, June 28, 2024).

### 3.2 Macronutrients

The overall nutrient distribution is shown as vertical profiles in Fig. 3 and as sea surface concentration plots in Fig. S 2. In surface waters, nutrient concentrations were uniformly low across the shelf. NOx concentrations were particularly low ($< LOQ = 0.21$ µmol L$^{-1}$) in surface waters and shallow shelf waters. Exceptions to this trend were observed at station 17 in Disko Bay and at stations 14, 15, 24 and 30 towards the west of each transect, where surface waters with higher nutrient concentrations were present (refer to Fig. S 2). Below the biological productive zone, nutrient concentrations gradually increased with depth, reaching maximum concentrations at station 15 (667 m) (refer to Table 1). Sherwood et al. (2021) reported similarly high nutrient concentrations in these deep waters, which are further discussed in Sect. 4.1.2.

In comparison to data collected during 2015 (GEOTRACES Intermediate Data Product Group, 2023) and 2016 (Bruyant et al., 2022), maximum nutrient concentrations of this study are lower, which we attribute to regional differences in the study areas, particularly with respect to the maximum sampling depth. We measured maximum values at the shelf edge at 667 m, whereas the referenced studies reported maximum values at stations located in central Baffin Bay, where sampling extended to significantly greater depths.

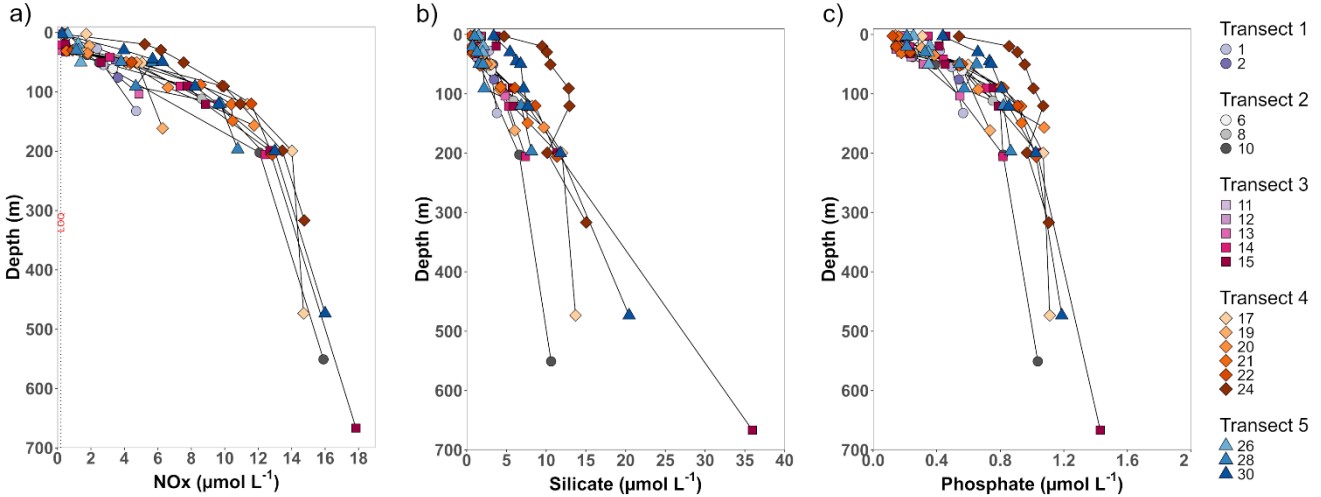

**Figure 3 Vertical profiles of nutrient concentrations across the study area. Each data point is identified by shape (transect number) and color (station number).**

### 3.3 Dissolved trace elements

The distributions of trace elements are shown as vertical profiles in Fig. 4 and surface water concentrations in Fig. S 3. The vertical profiles of dV and dCu showed opposing trends (Fig. 4 a and f). With depth, dV concentrations increased, while dCu concentrations decreased. The dV concentrations ranged from $1270 \pm 30$ ng L$^{-1}$ (station 24; 3 m) to $1810 \pm 60$ ng L$^{-1}$ (station 10; 551 m), while dCu concentrations ranged from $206 \pm 3$ ng L$^{-1}$ (station 24; 3 m) to $85.4 \pm 2.5$ ng L$^{-1}$ (station 10; 551 m), both matching minimum and maximum occurrences of salinity. Surface waters close to the coast of Greenland and at the western end of each transect showed lower dV concentrations (Fig. S 3 a) and higher dCu concentrations (Fig. S 3 f),

respectively. The distributions of dFe, dNi and dCd showed low concentrations in surface and subsurface waters, which gradually increased with depth (Fig. 4 c, e, and g; Fig. S 3 c, e, and g). Maximum concentrations were observed at depth, with dFe reaching $630 \pm 30$ ng L$^{-1}$ (station 18; 256 m), dNi reaching $307 \pm 9$ ng L$^{-1}$ (station 15; 667 m) and dCd reaching $52.2 \pm 1.2$ ng L$^{-1}$ (station 15; 667 m). Minimum concentrations of dFe, dNi and dCd were present in surface waters (refer to Table 1), particularly at stations with some distance to the coast. At several stations, dFe concentrations in surface waters were below the LOQ ($< 50$ ng L$^{-1}$). Surface concentrations of dFe and dNi (Fig. S 3 c and e) were elevated near the coast, decreased with increasing distance along each transect and increased again towards the west of each transect. In contrast, surface dCd concentrations were rather uniform across the shelf, with a slight increase toward the westernmost stations (Fig. S 3 g). The concentrations of dMn and dCo decreased from surface to subsurface waters and increased again with depth at stations located on the shelf (Fig. 4 b and d; Fig. S 3 b and d). In contrast, at the shelf edge, dMn and dCo concentrations remained stable with depth, resulting in minimum values of $54.8 \pm 1.8$ ng L$^{-1}$ for dMn (station 24; 317 m) and of $3.81 \pm 0.15$ ng L$^{-1}$ for dCo (station 10; 551 m). Elevated surface concentrations of both dMn and dCo were observed at station 17 in Disko Bay and at stations 11 and 12, located near the mouth of the Nassuttooq Fjord (Fig. 4 b and d; Fig. S 3 b and d). Surface concentrations of dMn and dCo generally decreased with increasing distance from the coast. The depth profile of dPb did not exhibit a consistent vertical pattern (Fig. 4 h). South of Davis Strait, elevated dPb concentrations were observed across transect 2 (refer to Fig. S 3 h), including a maximum dPb concentration of $4.04 \pm 0.13$ ng L$^{-1}$ in the deep waters of station 10. In contrast, north of Davis Strait, dPb concentrations were generally lower, falling below the LOQ ($< 0.98$ ng L$^{-1}$) in the deep waters of station 15 and the intermediate to deep waters of Disko Bay (stations 17 and 18). The minimum dPb concentration of $0.99 \pm 0.08$ ng L$^{-1}$ was observed in the deep waters of station 28.

Compared to literature values, our results for trace element concentrations are generally consistent with data collected during the 2015 Geotraces cruise GN02 (GEOTRACES Intermediate Data Product Group, 2023). Variations of minimum and maximum values are attributed to differences in sampling locations. While the present study took place on the shelf, capturing the influence of coastal runoff and sea ice meltwater, the Geotraces cruise focused on offshore stations located further north in central Baffin Bay.

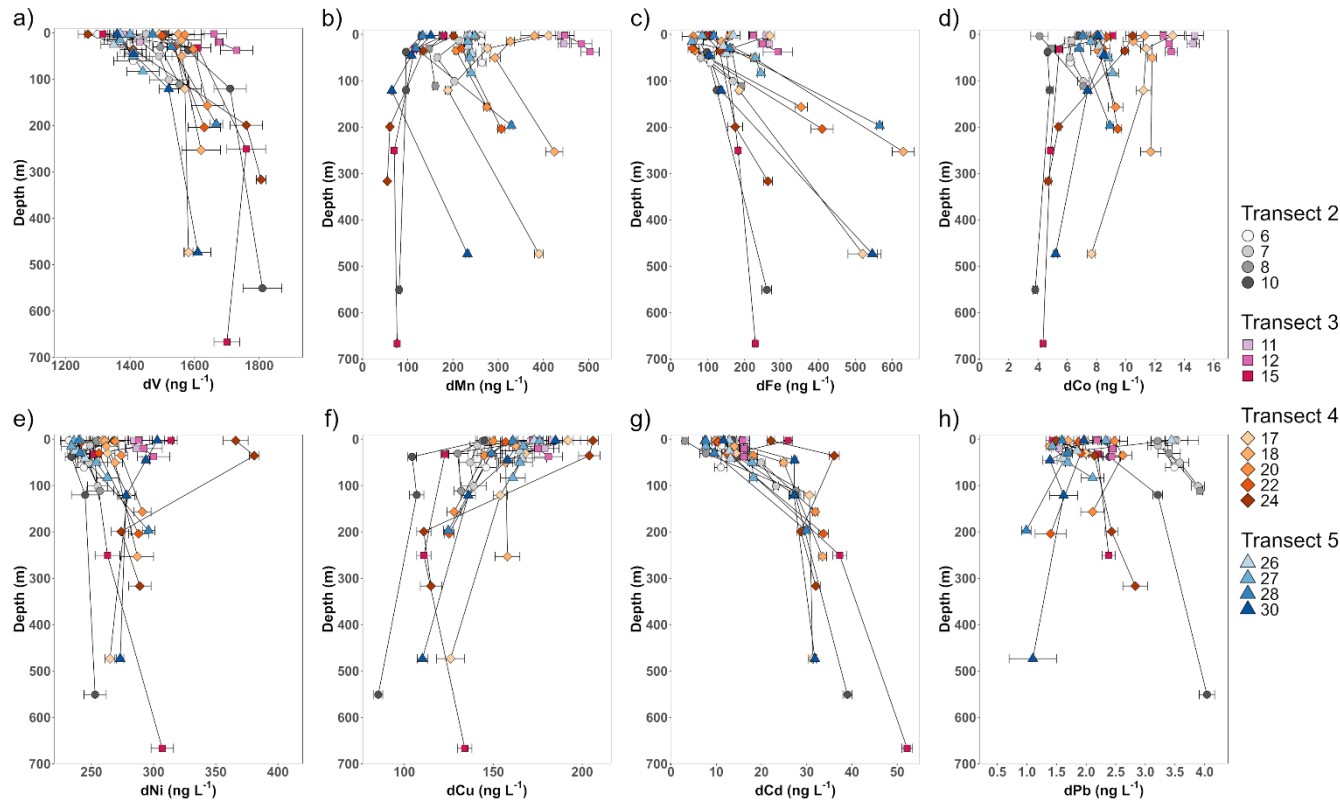

**Figure 4 Vertical profiles of trace element concentrations across the study area. Each data point is identified by shape (transect number) and color (station number). Error bars correspond to 2 SD.**

### 3.4 Carbonate system parameters

The vertical profiles of AT and CT (Fig. 5 a and b) show a gradual increase with depth. Minimum concentrations of AT and CT were present in the low-salinity surface waters at station 15 (AT: 2062 μmol kg$^{-1}$; CT: 1949 μmol kg$^{-1}$). While maximum CT concentrations of 2214 μmol kg$^{-1}$ were observed in the deep waters of station 15, AT concentrations followed the salinity gradient, reaching a maximum value of 2305 μmol kg$^{-1}$ in the deep waters of station 10. The vertical profiles of pH and $p$CO$_2$ (Fig. 5 d and e) show opposing trends, with pH increasing and $p$CO$_2$ decreasing towards intermediate waters of the photic zone. Below the photic zone, pH gradually decreases and $p$CO$_2$ increases with depth. This trend results in a pH maximum (8.20) and a $p$CO$_2$ minimum (249 μatm) in the photic zone of station 15 at 32 m, while the deep waters of the same station show a pH minimum (7.86) and a $p$CO$_2$ maximum of (559 μatm). The vertical profiles of the CT:AT ratio, Revelle factor and $\Omega$ Aragonite (Fig. 5 c, f and g) show that these parameters remain relatively stable within the upper 25 m of the water column. Below this depth, the CT:AT ratio and Revelle factor gradually increase, while $\Omega$ Aragonite decreases. The maximum CT:AT ratio (0.968) was observed in the deep waters of station 15, coinciding with a maximum Revelle factor of 17.80 and a minimum $\Omega$ Aragonite value of 0.90.

The surface water concentrations of the carbonate system parameters are given in Fig. S 4. In general, stations closest to the Greenland coast exhibited lower AT and CT values, which increased with distance to the coast. Following this trend, surface water pH increased and $p\mathrm{CO_2}$ values decreased along each transect with increasing distance from the coast. In coastal surface waters at stations 6 and 11, higher CT:AT ratios (0.920 and 0.925) coincided with higher Revelle factors (12.8 and 13.3) and lower $\Omega$ Aragonite values (1.93 and 1.82), indicating a reduced buffering capacity in coastal waters. In contrast, surface waters in Disko Bay (station 17; 3 m) exhibited a lower CT:AT ratio (0.907), driven by a low CT concentration of 1972 µmol kg$^{-1}$ and a corresponding low $p\mathrm{CO_2}$ value of 255 µatm. This was accompanied by a high pH (8.19), a low Revelle factor (11.85), and a maximum $\Omega$ Aragonite value of 2.17, indicating the higher buffering capacity of this water. In the westernmost surface waters of each transect, AT and CT values were again lower, but CT:AT ratios were among the highest observed. As a result, $p\mathrm{CO_2}$ values and Revelle factors were elevated at these stations, while pH and $\Omega$ Aragonite values were reduced.

Compared to AT and CT data collected in 2015 (H. Thomas, personal communication, June 28, 2024) and 2016 (Miller et al., 2020), our results fall within a similar range. However, the minimum values observed in this study are slightly lower, likely due to the influence of sea ice meltwater, which was captured during our sampling.

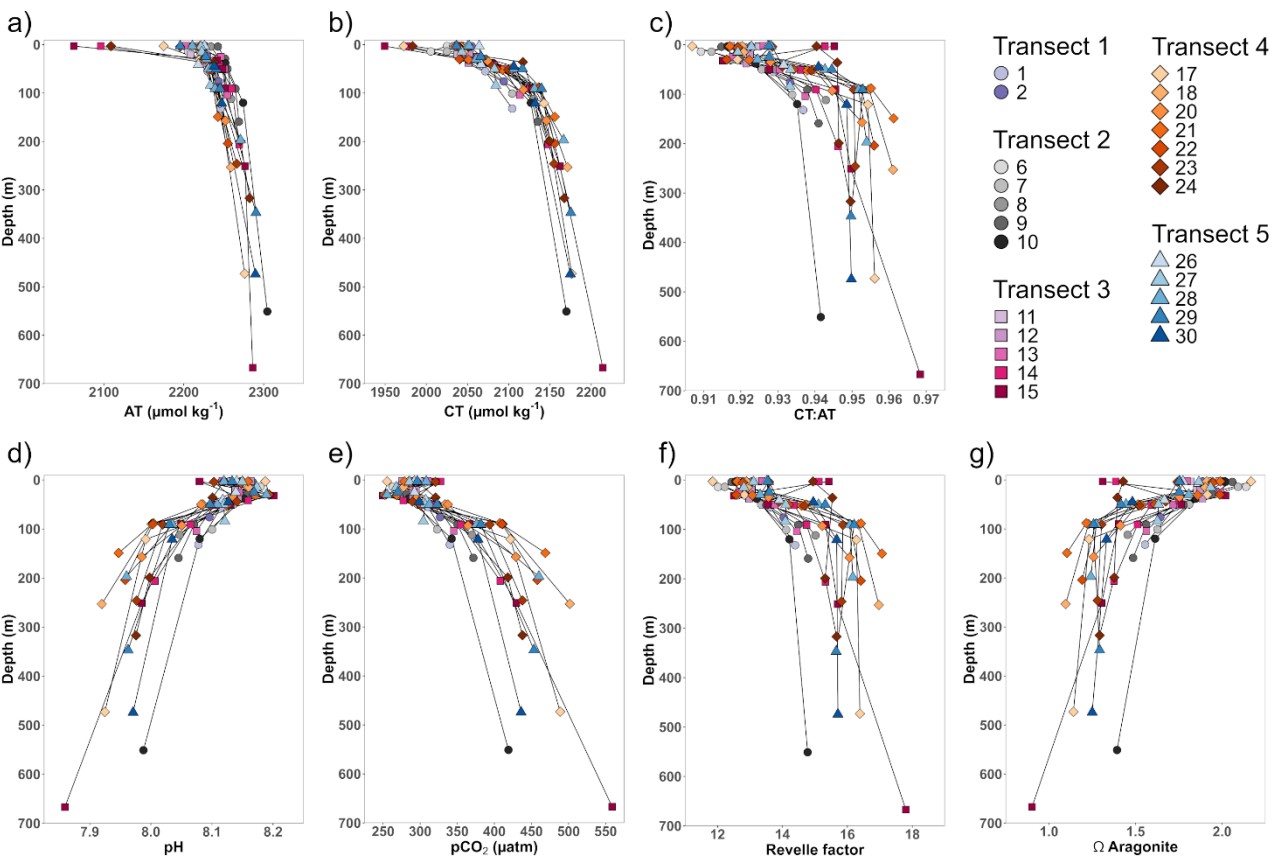

**Figure 5 Vertical profiles of carbonate system parameters across the study area. Each data point is identified by shape (transect number) and color (station number).**

### 3.5 Characterization of water masses

The distribution of water masses along the west coast of Greenland has been the subject of considerable discussion over the past decades, leading to the development of multiple classification systems and a wide range of nomenclatures. In this study, we identify water masses based on characteristic salinity and potential temperature ranges, as illustrated in Fig. 6 (black boxes; based on Tang et al., 2004; Curry et al., 2011, 2014; Sherwood et al., 2021). This terminology is applied consistently throughout the discussion. For comparison, a more recent interpretation of water mass structure off the west Greenland coast, as proposed by Rysgaard et al. (2020), is also included in Fig. 6 (orange labels). The West Greenland Shelf Water (WGSW; $\Theta < 7°C$; $S < 34.1$) was the dominant water mass in the study area, occurring in each transect across the shelf above a water depth of 200 m. The WGSW is also referred to as Southwest Greenland Coastal Water (CW), originating from the East Greenland Current and modified by freshwater input from Greenlandic glacial runoff. The West Greenland Intermediate Water (WGIW; $\Theta > 2°C$; $S > 34.1$) was present in deeper waters along the slope below 200 m. This water mass corresponds to the Subpolar Mode Water (SPMW), of which we identified a colder and less saline variant – the deep Subpolar Mode Water (dSPMW) – along the slope. Arctic Water (AW; $\Theta \leq 2°C$; $S \leq 33.7$) was detected north of Davis Strait at transects 3, 4, and 5, forming an intermediate layer between WGSW and WGIW above 100 m on the western side of the transects. These cold, low-salinity AW pockets can be observed in Fig. 2 h, k, and n. At transect 4, AW was found at station 17 approximately 70 km off the coast of Greenland, extending into Disko Bay. The updated classification refers to this water mass as Baffin Bay Polar Water (BBPW), of which a diluted form was found west of transects 4 and 5. The Transitional Water (TrW; $\Theta \leq 2°C$; $S > 33.7$) was primarily observed at transects 4 and 5 between 100 and 200 m, separating AW from WGIW. At station 15, the waters below 300 m were defined as Baffin Bay Water (BBW; $\Theta \leq 2$, $S > 34.1$), a shallower extension of the Baffin Bay Deep Water, presenting as a distinct tail on the $\Theta$-S diagram (Sherwood et al., 2021). Due to very low surface salinity values at stations 14, 15, and 24 ($S < 31.5$), we introduced sea ice meltwater (SIM) as an additional source water type to distinguish areas affected by sea ice melt.

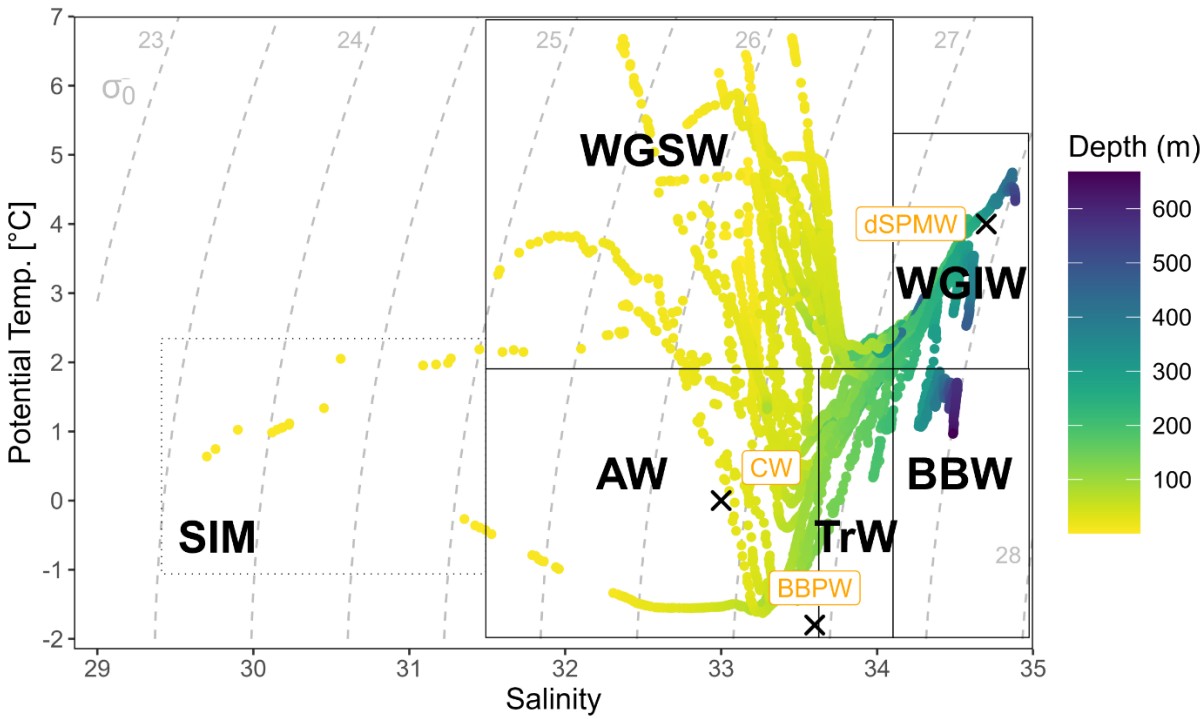

**Figure 6: Water masses in the study area were characterized using potential temperature and salinity: West Greenland Shelf Water (WGSW; Θ < 7°C; S < 34.1), West Greenland Intermediate Water (WGIW; Θ > 2°C; S > 34.1), Transitional Water (TrW; Θ ≤ 2°C; S > 33.7), Arctic Water (AW; Θ ≤ 2°C; S ≤ 33.7) and Baffin Bay Water (BBW; Θ ≤ 2°C, S > 34.1), according to Tang *et al.* (2004), Curry *et al.* (2011), Curry *et al.* (2014), and Sherwood *et al.* (2021). Surface water with S < 31.5 is indicative of sea ice meltwater (SIM). An updated perspective on the water masses was provided by Rysgaard et al. (2020): Baffin Bay Polar Water (BBPW; Θ = -**
**1.8°C, S = 33.6), Southwest Greenland Coastal Water (CW; Θ = 0°C, S = 33), and deep Subpolar Mode Water (dSPMW; Θ = 4°C, S = 34.7).**

## 4 Discussion

### 4.1 Influence of physicochemical processes on biogeochemical properties

The extensive data set was processed using PCA to find physicochemical processes that alter the distribution of carbon,
macronutrients, and trace elements. The results of the PCA analysis are summarized in Table S 6, where significant PC loadings of each parameter are highlighted in bold. Overall, 82 % of the total variance in the normalized data set can be explained by three PCs. The majority of variance is explained by PC 1 with 52 %, followed by PC 2 with 18 % and PC 3 with 12 %. Salinity, depth, oxygen, AT, CT, dV, and dCu significantly load on PC 1; potential temperature, AOU, NOx, silicate, phosphate, dNi, dCd, and dPb significantly load on PC 2; and dFe, dMn, dCo, dNi, and dCu significantly load on PC 3. In
Fig. 7, the results of the PCA as biplots with colors indicating the water mass affiliation of each sample are provided.

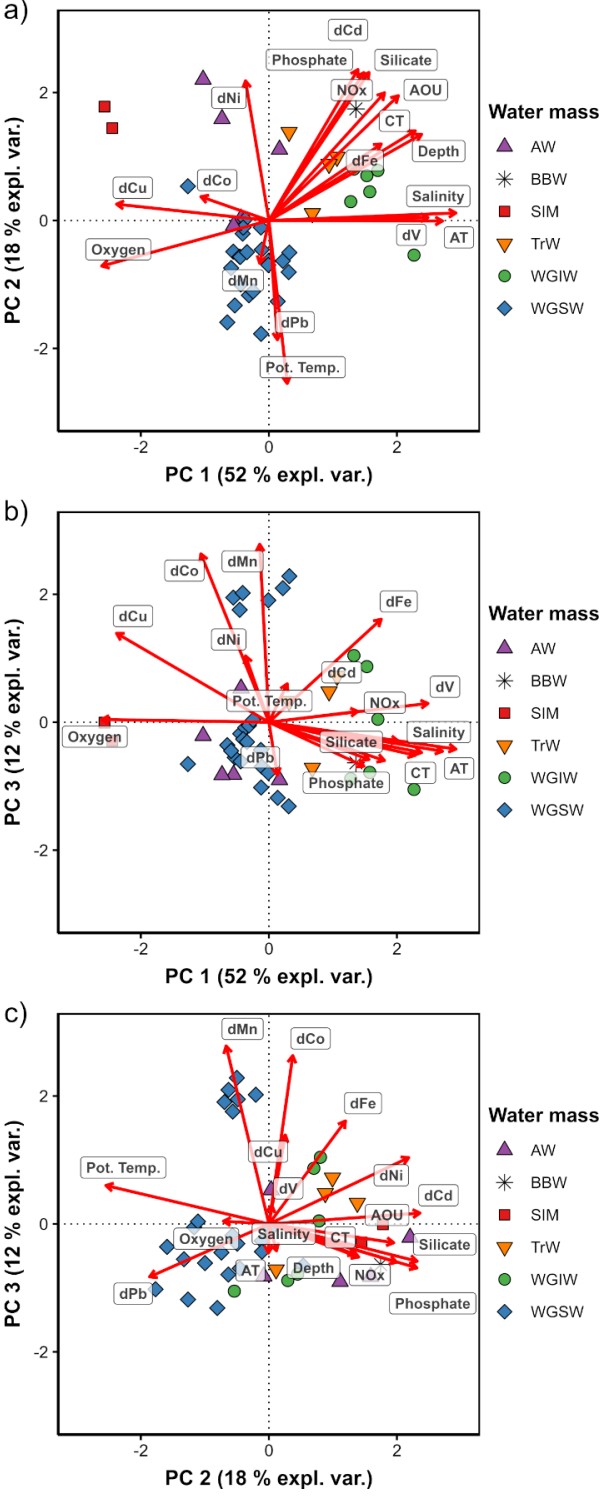

**Figure 7: Biplots according to PCA results for a) PC 1 score vs PC 2 score; b) PC 1 score vs PC 3 score; and c) PC 2 score vs PC 3 score, with colors indicating the water mass affiliation of each sample. The loading of each parameter is indicated by a red arrow.**

### 4.1.1 Conservative mixing along the salinity gradient (PC 1)

The comparison of low PC 1 scores in surface water samples with lower salinity (WGSW, AW, and SIM) and high PC 1 scores in deeper, more saline water samples (WGIW, TrW, and BBW) indicates that PC 1 resolves the salinity gradient of the study area (Fig. 7 a and b). The freshwater character of WGSW, AW, and SIM is clearly reflected in their low PC1 scores. The WGSW integrates freshwater contributions from the GIS (Foukal and Pickart, 2023), while AW represents a mixture of winter-cooled water entering Baffin Bay on the eastern side of Davis Strait and cold, fresh Arctic Ocean water of Pacific origin flowing through the CAA (Curry et al., 2014). SIM contributes to the formation of a low-salinity surface layer that remains concentrated in the upper part of the water column (Haine et al., 2015).

The statistical results illustrate the importance of freshwater input on the chemical properties of the study area. The high loadings of AT, CT, dV, and dCu on PC 1 indicate that these parameters behave conservatively, with freshwater being either a source (dCu) or a cause for dilution (AT, CT, and dV) of parameter concentrations. The positive loadings of AT and CT on PC 1 are consistent with the general understanding that freshwater inputs from sea ice and glacial melt are characterized by low AT and CT concentrations (Bates and Mathis, 2009; Chierici and Fransson, 2009; Fransson et al., 2015). The positive correlations of AT and CT against salinity are illustrated in Fig. S 5 a and b. Similarly, the distribution of dV shows a positive correlation with salinity (Fig. S 5 c), as both sea ice melt and coastal freshwater inputs are associated with low dV concentrations (Marsay et al., 2018; Whitmore et al., 2019). The distribution of dCu correlated negatively with salinity and was primarily influenced by coastal freshwater input (Fig. S 5 d). Sea ice meltwater also contributed to elevated dCu concentrations in surface waters (Fig. S 3 f), particularly toward the western ends of Transects 4 and 5. Cu-binding ligands play an important role in maintaining Cu in the dissolved phase (Ruacho et al., 2022), thereby facilitating the transport of dCu across the Arctic (Arnone et al., 2023). This is reflected in the consistent lateral decrease of dCu concentrations away from the coastal sources along the shelf transects (refer to Sect. 3.3). The influence of freshwater input on the chemical properties of the water column is further discussed in Sect. 4.3, which focuses on sea ice meltwater, and in Sect. 4.4, which addresses coastal runoff.

### 4.1.2 Biological uptake and recycling (PC 2)

The high positive loading of AOU, NOx, silicate, phosphate, dNi, and dCd on PC 2 suggests that PC 2 describes distributions driven by biological cycling. The nutrient-type behavior of dNi and dCd, as discussed in Sect. 3.3, aligns well with the PCA results. Additionally, dFe also shows a high loading on PC 2, reflecting its nutrient-type character. This observation is consistent with previously published data for the Nansen Basin and the Barents Sea (Gerringa et al., 2021), the Chukchi Sea (Vieira et al., 2019), the Fram Strait (Krisch et al., 2022a) and the Canadian Arctic (Colombo et al., 2020).

The anticorrelation between nutrients (positive PC 2 loading) and potential temperature (negative PC 2 loading) reflects the spatial distribution of these parameters across the study area. Low PC 2 scores of WGSW indicate that the warmer shelf waters are highly productive, resulting in low nutrient concentrations. In contrast, the colder deep waters of TrW, WGIW and BBW

exhibit high PC 2 scores, suggesting remineralization processes that return nutrients and trace elements to the water column. Similarly, AW and SIM water samples also show high PC 2 scores, characterized by higher nutrient concentrations alongside lower temperatures. The difference in nutrient concentrations between AW and WGSW is further explored in Sect. 4.2, while the influence of the retreating sea ice is discussed in Sect. 4.3.

The remineralization of organic matter (OM) led to elevated concentrations of NOx, silicate, phosphate, and dCd, along with maximum AOU values (refer to Sect. 3.1 and 3.2) in the deep waters of BBW (640 to 667 m; station 15). Although BBW was only encountered at this station, our results are consistent with Sherwood *et al.* (2021), who reported similar AOU and nutrient concentrations in BBW. The accumulation of remineralized nutrients at depth is attributed to the long residence time of deep and bottom waters in Baffin Bay, combined with the region's enclosed bathymetry, which restricts water circulation (Sherwood et al., 2021; Lehmann et al., 2019). In addition to nutrients, CT also shows a high loading on PC 2, whereas AT loads exclusively on PC 1. This distinction highlights the different biogeochemical behavior of AT and CT. While AT is primarily influenced by freshwater dynamics, CT is also significantly affected by biological processes. Consequently, elevated CT concentrations were observed in BBW (refer to Sect. 3.4), where restricted circulation and remineralization contribute to more corrosive conditions for Baffin Bay shelf sediments and benthic ecosystems. Similar to other studies in Baffin Bay (Beaupré-Laperrière et al., 2020; Burgers et al., 2024), OM respiration in deep waters led to undersaturation with respect to $\Omega$ Aragonite, accompanied by increased $p\text{CO}_2$ and decreased pH (refer to Sect. 3.4).

Alongside potential temperature, dPb exhibited a negative loading on PC 2, reflecting the latitudinal trend of potential temperature and dPb concentrations (refer to Sect. 3.1 and 3.3). The distribution of dPb is primarily controlled by the mixing of warm Atlantic-origin waters with high dPb signatures and cold Arctic-origin waters with low dPb signatures (Colombo et al., 2019). As a result, dPb concentrations were highest in the warmer waters southeast of Davis Strait and decreased progressively toward the colder northwest.

### 4.1.3 Scavenging and benthic processes (PC 3)

The third PC is associated with the trace elements dMn, dFe, dCo, dNi, and dCu. It is known that during estuarine mixing, dCo, dNi and dCu are co-cycled with dMn and dFe through their incorporation into (oxy)hydroxide mineral phases (Smrzka et al., 2019). We suggest that PC 3 indicates particle-driven mixing of these trace elements with coastal runoff as a source. High PC 3 scores (Fig. 7 b and c) of surface samples from stations 11 and 12 (mouth of Nassuttooq Fjord) as well as station 17 (Disko Bay) highlight the dominant influence of coastal freshwater inputs on the distribution of these trace elements. The role of coastal runoff in shaping trace element distribution is further discussed in Sect. 4.4.

The biplots (Fig. 7 b and c) reveal a notable feature of deep water samples (111 to 667 m), identified as TrW and WGIW. At these shelf stations, elevated concentrations of dMn, dFe, dCo, and dNi were observed in deep water samples (refer to Sect. 3.2), which we attribute to benthic processes occurring in surface sediments, i.e., reversible scavenging linked to mineral dissolution and the remineralization of organic material. Benthic release of dCo, dFe, and dMn has been documented in the Barents Sea (Gerringa et al., 2021), the Chukchi Sea (Vieira et al., 2019), the Canadian Arctic (Colombo et al., 2020) and

420 throughout the Arctic basin (Bundy et al., 2020), while benthic dNi release has been reported in the Bering Sea and Chukchi Sea (Jensen et al., 2022). In contrast, dCu did not exhibit signs of a benthic source or evidence of biological cycling. Instead, its distribution remained dominated by freshwater runoff from coastal sources, consistent with observations by Jensen *et al.* (2022).

A striking feature of these deep shelf water samples is the divergence in PC3 scores: some stations (8, 10, 15, 24, 30) show
negative scores, while others (17, 20, 22, 28) exhibit positive scores. We interpret this pattern as a reflection of spatial variability in return fluxes at the sediment-water interface. Stations with positive PC 3 scores are located closer to the coast, where greater export of biogenic and terrigenous material enhances recycling in deep waters. In contrast, offshore stations receive less particulate input, resulting in lower trace element concentrations and negative PC3 scores. A similar pattern was observed by Seo et al. (2022) in the East Sea, where significantly higher concentrations of dMn, dFe, and dCo were found on
the slopes and in the bottom layer of the Ulleung Basin compared to the Japan Basin. The authors attributed this to enhanced sedimentary release driven by the high OM content of shelf sediments (Seo et al., 2022).

### 4.2 Influence of major current systems on biogeochemical properties

The spatial distribution of parameters revealed significant differences in eastern and western shelf waters. To investigate these differences, we conducted a direct comparison of the three major water masses present along the west Greenland shelf: the
435 northward-moving WGSW (1 to 220 m) and WGIW (130 to 560 m) with the southward-moving AW (2 to 165 m). For each parameter, the median and the mean ± SD values are summarized in Table 2. The median is used throughout the data discussion because of its robustness against outliers.

**Table 2: Comparison of the median and mean ± SD concentrations between WGSW, WGIW and AW. The number of samples (*n*) used for the calculation is given in brackets.**

| | Parameter | WGSW (Median; Mean ± SD) | WGIW (Median; Mean ± SD) | AW (Median; Mean ± SD) |
|---|---|---|---|---|
| Basic oceanographic parameters (WGSW: $n = 3021$; WGIW: $n = 3034$; AW: $n = 2301$) | Salinity | 33.57 <br> 33.52 ± 0.42 | 34.45 <br> 34.45 ± 0.23 | 33.48 <br> 33.42 ± 0.27 |
| | Pot. Temp. (°C) | 2.68 <br> 3.20 ± 1.15 | 2.93 <br> 3.16 ± 0.73 | 0.38 <br> 0.15 ± 1.02 |
| | Oxygen ($\mu$mol L$^{-1}$) | 336 <br> 344 ± 23 | 273 <br> 278 ± 21 | 332 <br> 332 ± 23 |
| | AOU ($\mu$mol L$^{-1}$) | -4 <br> 0 ± 26 | 62 <br> 55 ± 23 | 36 <br> 30 ± 24 |
| Macronutrients (WGSW: $n = 54$; WGIW: $n = 9$; AW: $n = 29$) | NOx ($\mu$mol L$^{-1}$) | 1.7 <br> 2.4 ± 2.8 | 13.5 <br> 13.7 ± 1.7 | 5.7 <br> 6.0 ± 2.8 |
| | Silicate ($\mu$mol L$^{-1}$) | 1.5 <br> 2.0 ± 1.8 | 10.6 <br> 11.5 ± 4.1 | 4.2 <br> 5.2 ± 3.5 |
| | Phosphate ($\mu$mol L$^{-1}$) | 0.23 <br> 0.31 ± 0.18 | 1.02 <br> 0.99 ± 0.13 | 0.66 <br> 0.68 ± 0.21 |
| Carbonate system parameters (WGSW: $n = 50$; WGIW: $n = 12$; AW: $n = 25$) | AT ($\mu$mol kg$^{-1}$) | 2226 <br> 2228 ± 19 | 2275 <br> 2278 ± 11 | 2237 <br> 2237 ± 6 |
| | CT ($\mu$mol kg$^{-1}$) | 2051 <br> 2056 ± 28 | 2164 <br> 2159 ± 16 | 2104 <br> 2104 ± 27 |
| | CT:AT | 0.922 <br> 0.923 ± 0.007 | 0.950 <br> 0.948 ± 0.006 | 0.941 <br> 0.941 ± 0.010 |
| | pH | 8.14 <br> 8.14 ± 0.03 | 7.99 <br> 7.99 ± 0.04 | 8.10 <br> 8.09 ± 0.06 |
| | $p$CO$_2$ ($\mu$atm) | 292 <br> 295 ± 22 | 433 <br> 425 ± 37 | 321 <br> 332 ± 52 |
| | $\Omega$ Aragonite | 1.88 <br> 1.86 ± 0.15 | 1.30 <br> 1.34 ± 0.12 | 1.48 <br> 1.49 ± 0.20 |

**Continuation of Table 2: Comparison of the median and mean ± SD concentrations between WGSW, WGIW and AW. The number of samples (*n*) used for the calculation is given in brackets.**

| | Parameter | WGSW (Median; Mean ± SD) | WGIW (Median; Mean ± SD) | AW (Median; Mean ± SD) |
|---|---|---|---|---|
| Trace elements (WGSW: $n = 31$; WGIW: $n = 8$; AW: $n = 8$) | dV (ng L$^{-1}$) | 1490<br>1486 ± 234 | 1735<br>1713 ± 121 | 1480<br>1478 ± 127 |
| | dFe (ng L$^{-1}$) | 116<br>138 ± 67 | 262<br>329 ± 55 | 162<br>174 ± 26 |
| | dMn (ng L$^{-1}$) | 237<br>261 ± 71 | 89<br>164 ± 13 | 186<br>184 ± 20 |
| | dCo (ng L$^{-1}$) | 8.0<br>8.5 ± 2.5 | 5.0<br>5.7 ± 0.5 | 9.0<br>9.2 ± 0.8 |
| | dNi (ng L$^{-1}$) | 254<br>258 ± 51 | 269<br>270 ± 22 | 266<br>280 ± 20 |
| | dCu (ng L$^{-1}$) | 161<br>158 ± 44 | 111<br>111 ± 13 | 160<br>162 ± 16 |
| | dCd (ng L$^{-1}$) | 13<br>13 ± 6 | 31.6<br>32.2 ± 2.7 | 21.6<br>22.0 ± 2.3 |
| | dPb (ng L$^{-1}$) | 2.4<br>2.5 ± 0.8 | 2.4<br>2.4 ± 0.5 | 1.7<br>1.8 ± 0.4 |

The median values of salinity and potential temperature reflect the origin of the water masses and the freshwater dynamics of the area. In the upper water column, AW is slightly fresher (33.48) and colder (0.38°C) than WGSW (33.57; 2.68°C). This east–west gradient in temperature and salinity has also been described by Tang *et al.* (2004) between Baffin Island and the Greenland coast. As the warmer WGSW flows cyclonically around Baffin Bay and mixes with adjacent waters from the CAA, it becomes progressively colder and fresher, eventually returning southward as AW (Tang et al., 2004). The WGIW is part of the deep water WGSC that transports warm (2.93°C) and salty (34.45) waters from the North Atlantic (Curry et al., 2014). The AOU concentrations are reflective of respiration and the biological activity of the water (Sarmiento and Gruber, 2006). As AOU is a depth-dependent variable, we examined a consistent depth range of 30 to 50 m to enable a more meaningful comparison between WGSW and AW. Within this range, the median AOU values are -6 µmol L$^{-1}$ for WGSW and 7 µmol L$^{-1}$ for AW. This suggests that warmer shelf waters (WGSW) are more biologically productive than colder AW, as indicated by greater oxygen production in the upper water column. Although primary production was not directly measured in this study, this interpretation is supported by Krawczyk et al. (2021), who reported the highest chlorophyll *a* concentrations along the

southwest Greenland coast and the lowest values along Baffin Island. This pattern is consistent with overall lower nutrient concentrations observed in WGSW compared to AW (refer to Sect. 4.1.2 and Table 2), as nutrients are removed from the water column during PP. In Fig. 8 a, NOx concentrations are plotted against longitude within the 30 to 50 m depth range, including literature data from the Green Edge cruise in 2016 (Bruyant et al., 2022). A clear spatial gradient is evident, with lower NOx concentrations in WGSW on the shelf and higher NOx concentrations in AW further offshore, particularly in areas that were still ice-covered. This nutrient gradient reflects the east-to-west progression of sea ice retreat and has been linked to enhanced PP and species richness along the southern coast of west Greenland and in Disko Bay, compared to the more nutrient-rich but less productive regions near Baffin Island and northwest Greenland (Lafond et al., 2019; Krawczyk et al., 2021). We consider the timing of our sampling during the mid-summer season to be critical in capturing these pronounced differences in water mass composition. By this time, shelf waters had already become depleted in nutrients, whereas offshore regions were only beginning to experience increased biological processes following the retreat of sea ice (refer to Sect. 4.3). With increasing depth, respiration returns nutrients to the water column, as evidenced by high AOU values in WGIW (62 µmol L$^{-1}$) and overall higher nutrient concentrations at depth (refer to Table 2).

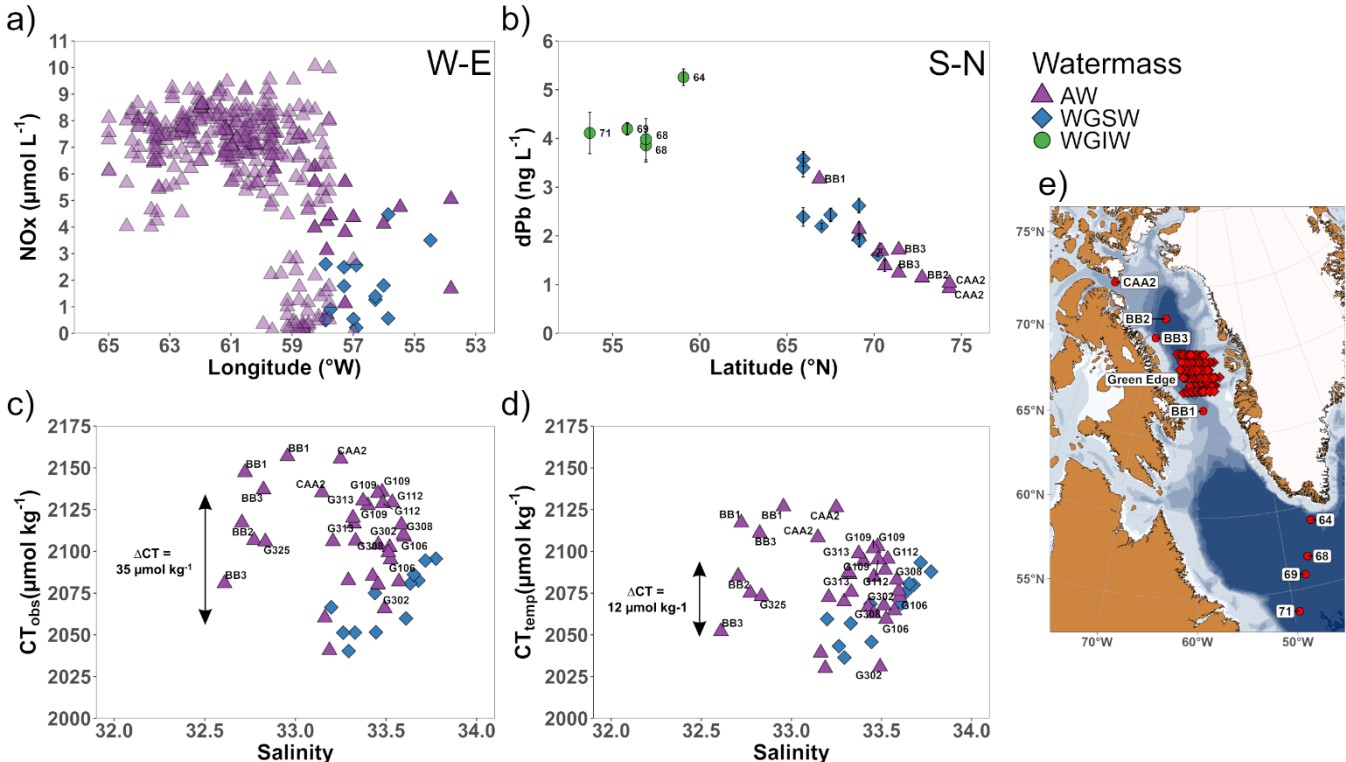

**Figure 8: a) NOx vs. longitude between 30 and 50 m with results of this study in solid colors and results of the Green Edge cruise in transparent colors (Bruyant et al., 2022); b) dPb vs. latitude between 30 and 50 m with results of this study (unlabeled) in comparison to results of the Geotraces GA01 cruise in 2014 (stations 64-71) and Geotraces GN02 in 2015 (BB1-BB3, CAA2), both: (GEOTRACES Intermediate Data Product Group, 2023); c) CT$_{obs}$ and d) CT$_{temp}$ vs. salinity between 30 and 50 m with results of this study (unlabeled) in comparison to results of the Green Edge cruise (Miller et al., 2020) and Geotraces GN02 in 2015 (BB1-BB3,**

CAA2) (H. Thomas, personal communication, June 28, 2024); e) location of the stations used as comparison. Different water mass categories are given in the legend.

A comparison of the carbonate system parameters in WGSW and AW reveals that both AT and CT median concentrations were lower in WGSW. This observation aligns with a study by Burgers *et al.* (2024), who reported that the dominant source of freshwater for WGSW in eastern Baffin Bay is glacial meltwater, characterized by a significantly lower AT endmember ($390 \pm 140$ µmol kg$^{-1}$) compared to AW in western Baffin Bay, where the AT endmember is $1170 \pm 120$ µmol kg$^{-1}$. The higher median CT:AT ratio in AW (0.941) compared to WGSW (0.922) is attributed to the CAA outflow, which carries Pacific-origin

water characterized by elevated CT concentrations (Azetsu-Scott et al., 2010; Shadwick et al., 2011b). In contrast, Atlantic-derived waters transported to eastern Baffin Bay tend to have lower CT:AT ratios (Burgers et al., 2024). These slightly higher CT:AT ratios in AW are also reflected in a higher median $p$CO$_2$ concentration, lower median pH value and lower median $\Omega$ Aragonite concentration (refer to Table 2).

    To minimize depth-related variability, we further examined CT values within a consistent depth range of 30 to 50 m. It is well

established that colder sea surface temperatures at high latitudes contribute significantly to the latitudinal gradient in surface CT, due to the increased CO$_2$ solubility of colder waters (Li and Tsui, 1971; Wu et al., 2019). To assess the extent to which temperature differences between AW and WGSW influence CT concentrations, we calculated temperature-normalized CT (CT$_{temp}$) following the method described by Wu et al. (2019) (see Sect. 2.4). This normalization was based on the median potential temperature of WGSW (3.23 °C, $n = 465$) within the 30 to 50 m depth range. The observed CT (CT$_{obs}$) and CT$_{temp}$

values for AW and WGSW are shown in Fig. 8 c and d. For comparison, we included literature data from the Green Edge cruise (Miller et al., 2020) and Geotraces GN02 in 2015 (BB1-BB3, CAA2) (H. Thomas, personal communication, June 28, 2024), filtered to the same depth range of 30 to 50 m. Including literature data, the difference between the median CT values of AW and WGSW is $\Delta CT_{obs}^{WGSW-AW} = 35$ µmol kg$^{-1}$. After temperature normalization, the difference is significantly reduced to $\Delta CT_{temp}^{WGSW-AW} = 12$ µmol kg$^{-1}$. This indicates that the majority of the CT difference in AW and WGSW is driven by the

temperature difference between the two water masses. The remaining CT difference may be attributed to higher biological productivity in WGSW, which reduces CT concentrations, and to the mixing of AW with the CAA outflow, which is known to transport additional CT to Baffin Bay (Azetsu-Scott et al., 2010; Shadwick et al., 2011b; Burgers et al., 2024).

    In the deeper waters of WGIW, both AT and CT median concentrations increase, following the salinity gradient and reflecting the contribution of OM respiration to the release of CT to the water column. This is also evident in higher median $p$CO$_2$

concentrations and lower median $\Omega$ Aragonite concentrations in WGIW (refer to Table 2), consistent with observations by Burgers *et al.* (2024).

    All median concentrations of trace elements exhibiting a nutrient-type profile (dFe, dNi, and dCd) were lower in WGSW compared to AW. This difference is attributed to biological uptake during PP, which we found to be higher in WGSW. The return of these elements to the water column through the remineralization of OM is evident in the elevated median

concentrations observed in WGIW (refer to Table 2). In addition to biological cycling, variations in source water composition also contribute to differences in trace element concentrations. AW exhibited elevated concentrations for dFe, dMn, dCo, and

dNi (refer to Table 2), which are likely linked to the CAA outflow into Baffin Bay. This outflow has been shown to have higher dFe and dMn concentrations due to sediment resuspension in the benthic boundary layer (Colombo et al., 2020; Colombo et al., 2021), as well as higher dNi and dCu concentrations associated with Pacific-origin waters advected from the Canada Basin through the CAA (Jensen et al., 2022). In contrast to other trace elements, higher dPb median concentrations were observed in WGSW and WGIW compared to AW (refer to Table 2). As mentioned in Sect. 3.3, a latitudinal gradient in dPb concentrations exists across Davis Strait, with dPb concentrations decreasing along the west coast of Greenland. This trend is illustrated in Fig. 8 b, where dPb concentrations are plotted against latitude within the 30 to 50 m depth range. The figure combines our data with literature values from the Geotraces GA01 cruise in 2014 (stations 64 to 71) and Geotraces GN02 in 2015 (BB1-BB3, CAA2) (both: GEOTRACES Intermediate Data Product Group, 2023). The elevated dPb concentrations in WGSW and WGIW are likely influenced by freshwater discharge from the GIS (Krisch et al., 2022b) and the advection of North Atlantic waters (Colombo et al., 2019), both enriched in dPb. In contrast, AW exhibited much lower dPb concentrations, which we attribute to the influence of the CAA outflow, as this region is relatively isolated from anthropogenic sources of dPb (Colombo et al., 2019). Our results demonstrate that the distribution of dPb is primarily controlled by the mixing of water masses with distinct dPb signatures. As such, dPb can serve as a useful tracer for illustrating the gradual mixing and transformation of water masses along the west Greenland shelf.

## 4.3 Influence of retreating sea ice on biogeochemical properties

Baffin Bay and its surrounding areas are typically ice-free between July and October (Bi et al., 2019). Our statistical analysis highlighted the significant influence of SIM on the chemical composition of surface waters (refer to Sect. 4.1.1). To visualize the decline in sea ice concentration during the sampling period, we used the ASI sea ice concentration product provided by the University of Bremen (Spreen et al., 2008). In Fig. 9 a, the change in sea ice concentration between 19 July and 28 July 2021 is illustrated. At the beginning of our sampling period on 19 July 2021, two extensive areas with high sea ice concentration were present between 66°N-68°N and 70°N-74°N. By 28 July 2021. These areas had disappeared almost completely, indicating a rapid and complete sea ice retreat within just 10 days. This melt event significantly affected sea surface salinity and potential temperature, as shown in Fig. 9 b and c. Both parameters decreased towards the west along transects 3, 4 and 5, with salinity reaching a minimum of 29.70 (station 14; 3 m) and potential temperature of -0.36°C (station 24; 3 m). The category SIM (refer to Sect. 3.5) was identified in the upper 4 m of stations 14 and 24, and in the upper 8 m of station 15. Although not formally classified as SIM, a similar trend of reduced surface salinity and potential temperature values was observed at stations 10, 13, 23, 28, 29, and 30, adjacent to SIM stations (refer to Fig. 2). We suggest that this trend reflects the gradual advection of SIM from central Baffin Bay towards the southeast.

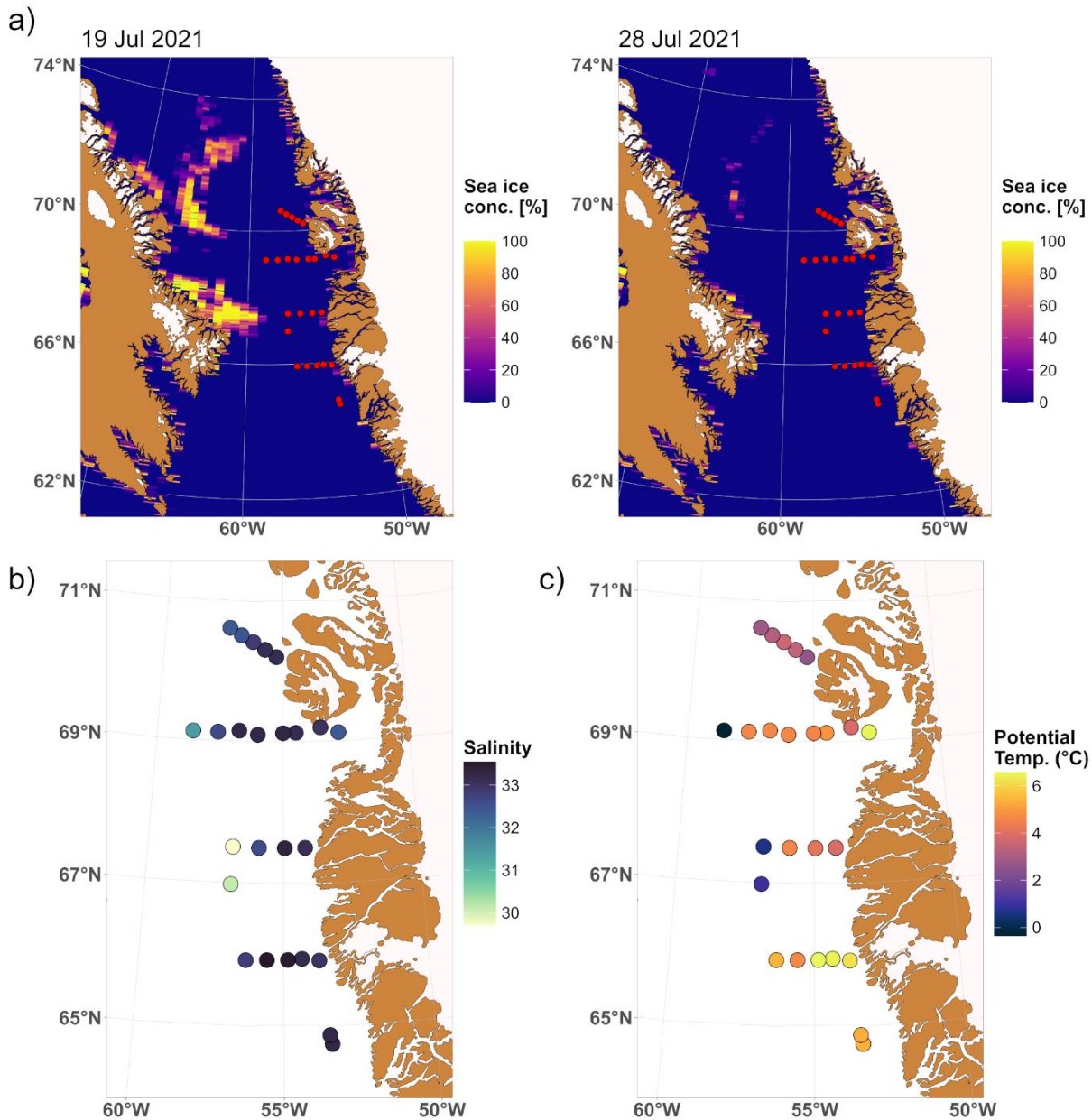

**Figure 9: a) Sea ice concentration (%) in the study area during the sampling period on 19 July and 28 July 2021. Sea ice concentration data (ASI data, ASMR2, version 5.4, 3.125 km grid) was downloaded from the data archive of the University of Bremen (Spreen et al., 2008). Sea surface b) salinity and c) potential temperature from CTD sensor data.**

In Baffin Bay, Lafond *et al.* (2019) observed elevated surface nutrient concentrations at stations covered by sea ice, in contrast to open-water stations located east of the ice edge, where nutrients were nearly depleted. In this study, the distribution of surface nutrients follows a similar trend (refer to Sect. 3.2). Along each transect, nutrient concentrations were consistently higher towards the west, coinciding with the recent retreat of sea ice. These western areas also exhibited elevated oxygen

concentrations and low AOU values (refer to Sect. 3.1), suggesting the onset of a phytoplankton bloom. The timing of the study enabled us to observe a transitional phase in the seasonal cycle: nutrients in the western shelf waters had not yet been fully consumed, while waters farther east had already become depleted due to earlier biological activity.

Our observation of the carbonate system indicates that surface waters of stations influenced by SIM exhibited lower AT and CT concentrations, along with higher CT:AT ratios. These conditions resulted in higher $p$CO$_2$ values and lower $\Omega$ Aragonite values (refer to Sect. 3.4). Similar results of sea ice meltwater increasing $p$CO$_2$ and decreasing CaCO$_3$ mineral saturation states have been reported along the east and west coast of Greenland (Henson et al., 2023), the Fram Strait (Tynan et al., 2016), and in other Arctic regions such as the Canada Basin (Bates et al., 2009). To investigate the effect of sea ice on the upper 50 m of the water column, which is the typical penetration depth of sea ice meltwater (Jones et al., 1983), we calculated salinity-normalized AT (AT$_{sal}$ = 35AT/S) and temperature-salinity-normalized CT (CT$_{temp}^{sal}$ = 35CT$_{temp}$/S), according to Wu $et$ $al.$ (2019) (refer to Sect. 2.4. and 4.2). These normalizations account for the effects of temperature and salinity on CO$_2$ solubility, while remaining sensitive to biological processes, air–sea gas exchange, and water mass mixing (Shadwick et al., 2011a). For AT$_{sal}$ (Fig. 10 a), most surface water concentrations ranged between 2330 and 2350 µmol kg$^{-1}$. We observed an AT addition of $\approx$ 25 to 55 µmol kg$^{-1}$ for SIM stations and of $\approx$ 5 to 45 µmol kg$^{-1}$ for SIM adjacent stations. These values are consistent with Jones $et$ $al.$ (1983), who reported an AT addition of $\approx$ 100 µmol kg$^{-1}$ further north in Baffin Bay, where the influence of sea ice meltwater is more pronounced. This AT enrichment is attributed to the release of alkalinity during the melting of sea ice that contains ikaite, as described by Rysgaard $et$ $al.$ (2011 and 2012) and Jones $et$ $al.$ (1983). The expected increase in CT by ikaite dissolution should be half the corresponding change in AT (Jones et al., 1983), or approximately 15 to 30 µmol kg$^{-1}$. The CT$_{temp}^{sal}$ values are shown in Fig. 10 b, with most surface water concentrations ranging between 2140 and 2170 µmol kg$^{-1}$. At SIM stations, we observed an increase in CT$_{temp}^{sal}$ of $\approx$ 60 to 75 µmol kg$^{-1}$, and at SIM adjacent stations of $\approx$ 10 to 60 µmol kg$^{-1}$. Hence, the observed CT addition exceeds the expected contribution from ikaite dissolution alone. Possible explanations for the observed surplus in CT include the release of CT from organic carbon remineralization under the ice (Bates et al., 2009; Tynan et al., 2016), or within the ice itself, as sea ice is known to contain higher concentrations in OM than the underlying seawater (Vancoppenolle et al., 2013). Additionally, the inflow of CT-enriched waters transported through the CAA into Baffin Bay may contribute to the elevated CT levels (Azetsu-Scott et al., 2010; Shadwick et al., 2011b; Henson et al., 2023). Our results indicate that the dissolution of ikaite in sea ice acted as a source of AT, providing a small geochemical buffer to surface waters (Jones et al., 2021). However, the inflow of Pacific-origin waters and remineralization under and within the ice contributed to elevated CT:AT ratios, resulting in higher $p$CO$_2$ and lower pH and $\Omega$ Aragonite values in surface waters. There is evidence that once the sea ice melt facilitated bloom fully develops and CT is removed from seawater by PP, higher pH and $\Omega$ Aragonite values can be expected along the sea ice edge (Tynan et al., 2016). This process is reflected in the photic zone of station 15 at 32 m, where an oxygen maximum and AOU minimum indicate high PP, resulting in a pH maximum (8.20) and $p$CO$_2$ minimum (249 µatm) (refer to Sect. 3.4).

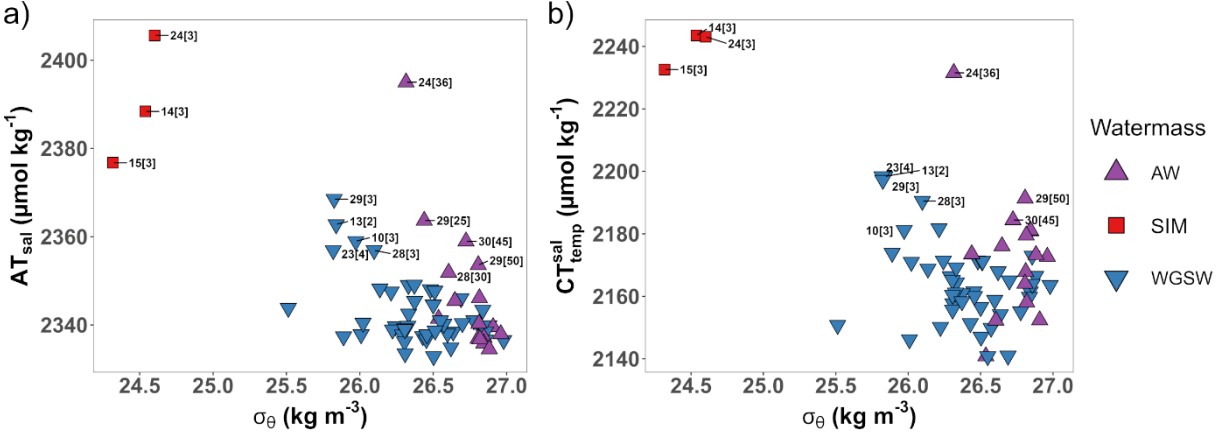

**Figure 10: First 50 m of the water column for a) salinity-normalized AT values against pot. density and b) temperature-salinity-normalized CT values against pot. density. Different water mass categories are given in the legend. SIM affected stations (14, 15, 24) and stations adjacent to those (10, 13, 23, 28, 29, 30) are labeled. The number in brackets corresponds to the sampling depth.**

We observed elevated surface water concentrations of dFe, dCo, dNi, dCu, and dCd in areas recently covered by sea ice, particularly towards the west of each transect (refer to Sect. 3.3). This observation aligns with a study by Tovar-Sánchez *et al.* (2010), who observed an enrichment of these trace elements in sea ice relative to surface waters, suggesting that sea ice meltwater is a significant source of these trace elements to surface waters in Baffin Bay. In particular, the additional input of dFe from melting sea ice may play a biologically important role in maintaining ice-edge blooms, as previously observed, e.g., in the Bering Sea (Aguilar-Islas et al., 2008). In addition to sea ice input, we suggest that, similar to macronutrients, trace elements had not yet been removed from surface waters by biological uptake at the time of sampling. We hypothesize that following the development of a bloom, concentrations of both macronutrients and trace elements would decline, as already observed in surface waters at shelf stations closer to the Greenlandic coast. Consistent with findings from other Arctic regions (Nakanowatari et al., 2007; Kanna et al., 2014), the progressive melting and retreat of sea ice in Baffin Bay and Davis Strait appears to significantly influence the biogeochemical cycling of carbon and trace elements, as well as the biological productivity of the upper water column.

**4.4 Influence of freshwater runoff from coastal sources on biogeochemical properties**

The influence of coastal freshwater runoff on the chemical composition of surface waters was demonstrated by statistical analysis (refer to Sect. 4.1.3) and is evident in minimum salinity values observed at stations 6, 11, and 17 (refer to Sect. 3.1). Stations 6 and 11 are located near to the mouths of the Kangerlussuaq Fjord and the Nassuttooq Fjord, both of which receive glacial and riverine freshwater from the GIS (Monteban et al., 2020). Station 17 is located in Disko Bay, approximately 70 km from the mouth of the Ilulissat Icefjord/Jakobshavn Isbræ system, where glacially modified waters form a buoyant coastal current outside the fjord (Beaird et al., 2017). These low-salinity surface waters coincide with elevated macronutrient concentrations (refer to Sect. 3.2). A modeling study by Møller *et al.* (2023) demonstrated that a portion of the macronutrient input into Disko Bay is directly linked to freshwater runoff. Additional mechanisms that transport marine-sourced nutrients

from deeper waters into the photic zone include vertical mixing fueled by wind and tide (Møller et al., 2023), as well as estuarine upwelling circulation forced by the glacier's freshwater input (Williams et al., 2021). Our results suggest that the continued replenishment of macronutrients into the photic zone of Disko Bay can sustain PP into the summer season. This is evidenced by high oxygen and low AOU concentrations in surface waters at station 17 (refer to Sect. 3.1). These conditions have significant implications for the carbonate system and the uptake of $CO_2$, as reflected by low Revelle factors and low $p$CO$_2$ values in surface waters of Disko Bay (refer to Sect. 3.4). A similar pattern was observed in the Godthåbsfjord system in southern Greenland, where biological processes were identified as the primary drivers of $CO_2$ uptake, resulting in low surface water $p$CO$_2$ (Meire et al., 2015). Based on these findings and supporting literature, we propose that nutrient cycling in Disko Bay is strongly influenced by GIS-derived freshwater input. This input stimulates PP and establishes Disko Bay as an important seasonal sink for atmospheric $CO_2$ well into the summer.

Apart from the Disko Bay area, surface waters at stations closest to the Greenland coast exhibited lower $\Omega$ Aragonite values with lower buffering capacities (refer to Sect. 3.4). Glacial freshwater discharge from the GIS is known to have a strong dilution effect on AT relative to CT, which reduces the buffering capacity of surface waters, resulting in lower $\Omega$ Aragonite saturation states and pH values (Henson et al., 2023; Burgers et al., 2024). However, surface water acidification can be mitigated by PP, which removes CT from the photic zone, consequently increasing both pH and $\Omega$ Aragonite (Chierici and Fransson, 2009), as observed in Disko Bay. In stations along the coast, surface waters exhibited negative AOU values, indicating CT drawdown via PP. Simultaneously, extremely low NOx values (< LOQ) were detected, suggesting that PP was approaching nutrient limitation. This nutrient depletion likely constrained further CT uptake, thereby no longer compensating the AT dilution effect caused by the GIS freshwater input. As a result, surface waters exhibited lower pH values coupled with higher Revelle factors and $p$CO$_2$ values. Our findings suggest that as the summer season progresses and PP declines, surface waters along the west Greenland coast become increasingly vulnerable to acidification due to the input of poorly buffered glacial freshwater. These observations are consistent with Henson *et al.* (2023), who observed that AT dilution from glacial meltwater drives corrosive conditions in Greenlandic coastal surface waters, leading to a general decline in both $\Omega$ Aragonite and pH (refer to Sect. 3.4).

In addition to influencing the cycling of macronutrients and carbon, coastal freshwater input significantly affects the distribution of trace elements. Surface water concentrations of dMn, dFe, dCo, dNi, and dCu were generally highest close to the coast and decreased with increasing distance along each transect (refer to Sect. 4.2.3). This trend was especially pronounced at stations 11 and 12 (mouth of Nassuttooq Fjord), and station 17 (Disko Bay, Ilulissat Icefjord). While surface water concentrations of dNi in Disko Bay remained relatively constant, elevated concentrations were observed at the mouth of Nassuttooq Fjord. Similar spatial variability in dNi has been reported by Krause *et al.* (2021), who related these trends to the presence of outcrops of Ni-rich minerals along the Greenland coast. Overall, our work is consistent with previous studies from various regions around Greenland, which identified GIS-derived freshwater as a source of trace elements including dMn, dFe, dCo, dNi, and dCu (Colombo et al., 2020; Hawkings et al., 2020; Krause et al., 2021; Chen et al., 2022; Krisch et al., 2022a;). The input of bioactive trace elements, e.g., dMn, dFe, and dCo, via GIS freshwater runoff has been suggested to enhance

primary productivity in adjacent shelf areas (Bhatia et al., 2013; Hawkings et al., 2020). Our study provides further evidence that freshwater from the GIS may support biological processes by supplying bioactive trace elements to surface waters.

**5 Summary and Conclusion**

The distribution of macronutrients, carbonate system species, and dissolved trace elements on the west Greenland shelf is
highly dynamic and influenced by the complex interplay between physicochemical drivers, major ocean currents, sea ice melting, and terrestrial freshwater runoff from the GIS.

The biogeochemical cycling of these parameters is governed by several key mechanisms. First, conservative mixing along the salinity gradient distinguishes fresher surface waters (WGSW, AW, SIM) from more saline deep waters (WGIW, TrW, BBW). While AT, CT, and dV show positive correlations with salinity due to dilution by freshwater, dCu shows a negative correlation,
indicating coastal and sea ice meltwater as sources. Second, biological processes regulate the vertical fluxes of macronutrients, CT, and trace metals such as dFe, dNi, and dCd by photosynthetic uptake and OM remineralization. In the deep waters of Baffin Bay, the long residence time and the enclosed bathymetry promote the accumulation of remineralized nutrients and CT. This accumulation drives corrosive conditions, resulting in $\Omega$ Aragonite undersaturation in deep waters of southern Baffin Bay. Such conditions pose a potential threat to the benthic ecosystem, particularly by creating an unfavorable environment for
shell-forming marine organisms. Third, benthic fluxes across the sediment-water interface contribute to elevated concentrations of certain trace elements (e.g., dMn, dFe, dCo, and dNi) in bottom waters of shelf stations. We hypothesize that the magnitude of this return flux decreases with increasing distance from the Greenlandic coast, likely due to reduced deposition of biogenic and terrigenous material available for remineralization and reversible scavenging. Further investigation of Baffin Bay sediments is necessary to quantify benthic fluxes of trace elements and geochemical reactions occurring within
the sediment. Our study provides an initial assessment of the role of benthic inputs in trace element cycling on the west Greenland shelf and Baffin Bay, highlighting their potential significance in shaping regional biogeochemistry.

The distribution of chemical constituents in the water column across the west Greenland shelf is primarily influenced by the opposing directions of the BIC and the WGC and their respective water masses. The BIC transports the southward-moving AW, which combines winter-cooled water from Baffin Bay with Pacific-origin Arctic Ocean water entering through the CAA.
We characterized this water mass as cold and fresh, with elevated nutrient concentrations and high CT:AT ratios, reflecting both the lower biological activity and the additional CT input from the CAA. Elevated concentrations of dFe, dMn, dCo, dNi, and dCu were observed in AW, likely due to the influence of trace element-rich outflows from the CAA into Baffin Bay. In contrast, the WGSW, transported northward by the WGC, integrates freshwater of Arctic-origin from East Greenland with glacial meltwater from the GIS. This water mass dominated the upper water column on the shelf and is characterized as warm
and fresh, with high biological productivity and low nutrient concentrations. Beneath the WGSW lies the WGIW, a warm and salty water mass of North Atlantic origin. This water mass is enriched in nutrients and carbon due to the remineralization of

OM. The gradual mixing of these three water masses along the shelf was further illustrated by the latitudinal distribution of dPb, which is influenced by elevated dPb waters from the North Atlantic and low dPb waters from the CAA.

Our study provides evidence that the progressive melting and retreat of sea ice alters the biological productivity and the biogeochemical cycling of carbon and trace elements in surface waters of southern Baffin Bay. The east-to-west retreat of sea ice during spring and summer created a significant gradient in nutrient concentrations. Nutrients were lowest in the highly productive shelf waters along the eastern side of the west Greenland coast, while higher concentrations were observed towards the west of southern Baffin Bay, where sea ice cover persisted for a longer duration. The timing of this study enabled us to capture the onset of a phytoplankton bloom in areas previously covered by sea ice, where nutrients had not yet been depleted. Additionally, sea ice meltwater provided additional AT to surface waters, likely through the dissolution of ikaite. However, this small geochemical buffer was offset by elevated CT concentrations, potentially derived from the remineralization of OM under and within the ice and the inflow of CT-rich Pacific-origin waters via the CAA to Baffin Bay. Overall, the combined influence of sea ice meltwater and the inflow of Pacific-origin waters reduced AT relative to CT, resulting in elevated $p$CO$_2$ and decreased pH and $\Omega$ Aragonite values in surface waters. As the ice-edge bloom progresses and CT is removed by PP, an increase in pH and $\Omega$ Aragonite values can be expected along the retreating ice margin. Furthermore, we identified sea ice meltwater as a source of bioactive trace elements, including dFe, dCo, dNi, dCu, and dCd to surface waters of southern Baffin Bay. This additional input of micronutrients may sustain and prolong the duration of ice-edge blooms, with implications for regional productivity and biogeochemical cycling.

Freshwater runoff from the GIS significantly influenced the chemical composition of coastal waters along west Greenland, particularly in Disko Bay and at the mouth of the Nassuttooq Fjord. In the Disko Bay area, GIS-derived freshwater input replenished macronutrients in the photic zone, stimulating PP and creating an important sink for CO$_2$ long into the summer season. The supply of bioactive trace elements via GIS freshwater runoff, could further support biological processes in the surrounding shelf areas. However, in coastal areas where PP approached nutrient limitation, surface waters became more susceptible to acidification due to the input of poorly buffered glacial freshwater.

This work successfully captured a high-resolution, large-scale snapshot of various water column parameters across the west Greenland shelf during July. Nevertheless, we were unable to resolve seasonal variability, which requires observations at higher temporal resolution. We emphasize the need for sustained time-series observations to fully assess the consequences of climate change in this climate-sensitive region.

## Data availability statement

The data set for the carbonate system and dissolved trace elements can be found online at: https://doi.org/10.5281/zenodo.14235091 (Schmidt et al., 2024).

The data set for the nutrients can be found online at: https://doi.org/10.5281/zenodo.15590652 (Stedmon et al., 2025).

## Supplement link

The supplements to this article can be found online at: XXX

## Author contribution

CES: Conceptualization, Formal Analysis, Investigation, Methodology, Data curation, Visualization, Writing – original draft

TZ: Writing – review & editing

KK: Investigation, Writing – review & editing

DP: Resources, Supervision, Writing – review & editing

HT: Funding acquisition, Project administration, Resources, Writing – review & editing

## Declaration of competing interest

The authors declare that they have no conflict of interest.

## Acknowledgement

The authors would like to thank the captain and crew of RV Dana for their support during the sampling campaign in 2021. Special thanks to Camilla Svensen and Ulrike Dietrich (both University of Tromsø) for the collection of the nutrient samples and Colin Stedmon and his team (Technical University of Denmark) for the nutrient analysis.

## Financial support

This project has received funding from the European Union's Horizon 2020 research and innovation program under Grant Agreement No. 869383 (ECOTIP) and the European Union's Horizon 2023 research and innovation program under Grant Agreement No. 101136480 (SEA-Quester). The project was made possible through the program "Changing Earth – Sustaining our Future" (Subtopic 4.1) within the Helmholtz Association.

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
