# Peer review of "Influences on Chemical Distribution Patterns across the west Greenland Shelf: The Roles of Ocean Currents, Sea Ice Melt, and Freshwater Runoff"

_EGUsphere, 2025_

## Referee Comment (RC2)

Review of Influences on Chemical Distribution Patterns across the West Greenland Shelf: The Roles of Ocean Currents, Sea Ice Melt, and Freshwater Runoff.

General comment. This work investigates inorganic carbon, macro and micronutrients along the West Greenland Coast, an area facing enhanced changes due to global warming, mainly enhanced glacial melt (both from glaciers and sea ice). It should be highlighted that it is not common to have such a wide spectrum of biogeochemical parameters, and to get insights from such an extensive set of parameters, the authors followed an original approach: PCAs. The sampling strategy is well adapted, as mentioned before, the number of parameters is impressive, the analysis and data treatment are also solid and state of the art, the PCA's approach is original and solid. The authors know the area and what they are doing. However, I have some relatively minor concerns.

Sometimes, the suggestions or conclusions are not always as strong as the wording suggests, so I would recommend some changes in the text. For example, I found the wording "corrosive", especially in line 563, inappropriate. I understand that it might appear necessary to have some attention-grabbing sentences/ideas, but the point is that if we consider that corrosive waters apply to waters with $\Omega_{aragonite}$ or $\Omega_{calcite}$ below 1, there are NO corrosive surface waters during that study (Figure S10). If there is no question that the glacial meltwater promotes the decrease of $\Omega_{aragonite}$ as it it shown by that study, we cannot draw any conclusion on the rate of the $\Omega_{aragonite}$ change, any perspectives of how it will change in the future and when. So I would refrain from this alarming tune. The work is interesting enough without having to oversell the conclusions, I believe.

There is evidence, well shown in the manuscript, of a higher CT:AT ratio in sea ice meltwater. The impact of ikaite is correctly addressed, and enhanced CT is well mentionned. This high CT is ascribed to enhanced organic carbon remineralisation below sea ice. Why not, but I suggest mentioning that this excess CT can also come from remineralisation within the ice itself. As it has been correctly mentioned elsewhere, there is an enhanced concentration of trace metals in sea ice. This enhancement comes from both the atmosphere and the scavenging of organic matter in sea ice during its growth. So, it is not only the trace metal concentration that is enhanced in sea ice, but also the organic matter concentration. This comes with enhanced respiration and release of CT within the ice. It could be stated in the manuscript in part 4.4.

Also, despite the very extensive set of parameters, I regret that there are many indirect assumptions about the productivity of the water masses, while a simple measurement of Chl a would have been probably more robust. I presume that if the information is not presented here, it is because it is unavailable. Otherwise, it will be more robust in my mind to show the phytoplanktonic biomass as well, to discuss the productivity of waters, especially at that time of the year.

Minor remarks

Line 169 *The calculation was based on the median potential temperature of WGSW (3.23°C) within a depth range of 30 to 50 m, using observed salinity and AT, and the in situ pCO2 as input parameters*. Is there any suggestion to chose that value as reference?

Line 235 "*The concentration of dFe and dNi in surface waters (Fig. S 9c and e) followed a similar pattern as the surface water nutrient concentrations (Fig. S 7)*. It is not clear to me to which pattern you are referring. I presume it is the decrease in concentration at the surface. But the pattern is not very clear to me, at least for dNi. For instance, it is much clearer for dCd, which

appears to me to have the clearest "nutrient-like" vertical profile. So, why are you mentioning specifically dNi and not dCd?

 3.4 Carbonate system parameters. Line 255 to 280. I have no concern about your general description of results, but my attention has been caught by the shift in CT:AT at the surface waters that might be addressed, especially station 23 (or 24), 14 and 05, with related impact on Revelle factor and aragonite saturation. I am wondering if you can address this striking/questioning feature.

Line 329 *"and sea ice 330 meltwater as a dCu source"*. While the source of Cu from glacial freshwater seems compelling from Figure S12, this is not the case for meltwater sea ice, as the value of dCu is the same at salinity 33 and 30.5-31. I acknowledge that we can expect sea ice meltwater from sea ice, according to current understanding of trace metals in Antarctic sea ice and Arctic sea ice (Tovar-Sánchez *et al.*, 2010). But this does not seem obvious to me in the current survey, at least from the figures S12, despite the PCAs.

Figure 4 a). I found it confusing to plot longitude from low to high, as this implies that the West is on the left and the East is on the right, especially as there is a map in the same figure, meaning that the East is on the left in fig 4e) and on the right in fig 4a.

Line 410. How robust is the comparison of AOU values in WGSW and AW? It is not clear to me if you are comparing similar depths. I believe this is not the case, and you are comparing the whole data set without considering the depth. I am not comfortable with that, as depth is one of the first predictors of AOU. In addition, the AOU is also affected by air-sea exchange, which depends on ice cover. So, in my opinion, the suggestion that warmer shelf waters being more productive than the colder AW, is not well supported by the comparison of AOU, even if I expect that is indeed correct.

Line 597: *"**Thus**, the benthic ecosystem of the west Greenland shelf is potentially vulnerable to future ocean acidification and suppression of CaCO$_3$ saturation states."* There is a bit of mixing, and confusion introduced there, in my opinion. There is only on single data below the aragonite saturation at 700m. This does not correspond to the condition on the shelf. What means "suppression of CaCO3 saturation states". Is it possible to "suppress" any saturation state? I am not a native English speaker, but the expression is a bit odd to me. I would have expected something like undersaturation in Aragonite, decrease in saturation state, but not a suppression.

Line 604: I appreciated the part "We hypothesize that this return flux decreased with increasing distance 605 from the Greenlandic coast as the amount of exported biogenic and terrigenous material available for remineralization and reversible scavenging also decreased. Further studies looking into Baffin Bay sediments are needed to disclose benthic fluxes of trace elements and reactions occurring therein. Our study provides a first insight into the importance of benthic inputs for trace element cycling on the west Greenland shelf and Baffin Bay." The evidence of the process, as derived from the PCAs is not fully compelling. I am not against the idea, but at that point, I appreciate that it is mentioned as only a suggestion.

Figure S3. Something is wrong with the plots. A large white square masks a large part of them. I don't know where the depth scale of the plot is.

I found the Table S5 odd, with values missing (max salinity station 15 to 30); Pot. Temp minimum station 8 to 17, and so on. In addition, I would have expected a single value as minimum or

maximum, rather than a range of values. Per se, a minimum value is only one value. The values are given for only some stations and not all. For example, there are no salinity values for 6 to 10, but there is a maximum potential temperature value for that station, but not a minimum value. It looks a bit random. Also, I'm wondering if the table would benefit from being rotated 90°. I mean, one line per station, and then each column corresponding to a parameter (min salinity, maximum salinity...., etc.).

Figure S1. This plot is made with only 2 stations parallel to the coast, while all other plots are based on transects of at least 5 stations perpendicular to the coast. I don't think that it is very useful to have it. And I'm sure the structures exhibited by the contour lines for salinity and oxygen have no physical meaning and can be removed.

Figure S9, S10 and others. I won't cut the symbol above the 0 depth. It makes the surface a little less easy to read. It looks like you applied a "blank" patch above the 0-depth line.

---

## Author Comment (AC1)

**Review of Influences on Chemical Distribution Patterns across the west Greenland Shelf: The Roles of Ocean Currents, Sea Ice Melt, and Freshwater Runoff**

The paper investigates the key drivers of chemical distribution patterns on the West Greenland shelf during summer. The study focuses on how ocean currents, sea ice melt, and terrestrial freshwater runoff affect the distribution of macronutrients, carbonate system parameters, and dissolved trace elements based on a cruise with CTD casts, water sampling for nutrients and carbonate chemistry, and trace element analysis. The study area spans across several transects along the Greenland shelf.

We sincerely thank the reviewer for the helpful and constructive comments. These suggestions have significantly improved the revised manuscript, particularly in terms of its overall structure and the presentation of the results. We greatly appreciate the time and effort the reviewer has dedicated to providing this valuable feedback.

While the research aims to provide valuable insights into this complex marine environment, the presentation of the study suffers from significant shortcomings in readability, structure, and data visualization. Sentences are often long, lacking a clear flow of ideas jumping between concepts without sufficient transition or explanation, especially in the result and discussion section.

Thank you for this feedback. We will reorganize the manuscript to improve its structure and readability, summarizing concepts in a more cohesive manner. For the revision, we will consult an external person who is a native English speaker to further enhance the wording and text structure.

Personally I feel the data is not presented effectively. While there are figures in the supplementary material, the main text lacks visuals to illustrate the actual data. Instead, it overuses tables, which don't show the data as clearly as figures would. Also a lot of new data and figures are presented in the discussion which makes readability harder.

In the results section, we will include a figure with multiple panels displaying various parameters (e.g. Sal, Temp, AOU, NOx, AT, CT, dMn, dFe, dCd, and dPb) for Transect 4. This figure will help visualize the data and introduce general concepts of the study area, highlighting similarities between the parameters (e.g. NOx and dCd, dMn and dCo).

Additionally, the referencing throughout the manuscript needs improvement. Several citations are inappropriate, not directly supporting the statements they accompany (See below as well). Key references relevant to the specific processes and region under study are also missing. For instance, the manuscript would benefit from including more recent and relevant work on the hydrography of the West Greenland shelf by Rysgaard et al. (2020)

We will correct the citation in the introduction as seen below. We will include the hydrography work by Rysgaard et al. (2020) in chapter 4.1 by comparing this updated view with the categorization system used throughout the rest of the manuscript. We will also reference other work using the same classification system (e.g. Mortensen 2022). The reason for using the older system was that it gives clear boundaries for the water masses that we used to describe and compare the chemical signatures throughout the discussion.

Below some specific comments

L20: sea ice retreat creates nutrient gradients: in general you have to be aware that would you present is just a snapshot of july, so be careful to make statements like this. Yes, there is a gradient in July, but not necessarily for August, September,..

Original sentence: The east-to-west direction of sea ice retreat creates nutrient gradients, with low nutrient levels in highly productive shelf waters and high nutrient levels in areas with prolonged ice cover.

Proposed changes:

➔ L15 "during late summer" change to "during July".
➔ During the study period, we were able to capture a distinct nutrient gradient following the east-to-west direction of sea ice retreat, with low nutrient levels in highly productive shelf waters and high nutrient levels in areas with prolonged ice cover.

L 33: This is not supported by paper you cite here AND L34-35: Look at newer circulation paper in West Greenland as well, eg Rysgaard et al. 2020

Original sentence: The circulation across the west Greenland shelf and Baffin Bay is dominated by the opposing direction of the Baffin Island Current (BIC) and West Greenland Current (WGC) (Curry et al., 2011).

Proposed changes:

➔ The circulation across the west Greenland shelf and Baffin Bay is dominated by two major currents in the region. The southward-flowing Baffin Island Current (BIC) along the Baffin Island continental slope and the northward-flowing West Greenland Current (WGC) along the west Greenland continental slope (Rysgaard et al., 2020).

L47: suggest to look at more appropriate referencing for sea-ice conditions

Krawczyk et al., 2021 use satellite-derived sea ice concentration as monthly average from 1978–2014, which is an appropriate reference. We will also include a reference that has a more global perspective and one that looks at sea ice in Baffin Bay in October and the changes of the melt period.

Original sentence: In the last two decades, sea ice concentrations in Baffin Bay and the Labrador Sea have decreased significantly due to climate change (Krawczyk et al., 2021).

Proposed changes:

➔ In the last two decades, sea ice concentrations in Baffin Bay and the Labrador Sea have decreased significantly due to climate change (Krawczyk et al., 2021), additionally driven by an earlier melt onset and later freeze-up (Stroeve and Notz, 2018; Ballinger et al., 2022).

L49: Do the studies provide evidence of this?

Yes, both do.

Original sentence: Reduced sea ice cover has been correlated with an overall increase in productivity and species richness along the west Greenland shelf (Krawczyk et al., 2021; Møller et al., 2023).

Proposed changes:

→ The reduction in sea ice cover, alongside the additional influx of Atlantic-sourced waters (Krawczyk et al., 2021) and increasing freshwater discharge (Møller et al., 2023), have been correlated with an overall increase in productivity and species richness along the west Greenland shelf (Krawczyk et al., 2021; Møller et al., 2023).

L55: in early-summer

Original sentence: This is why macronutrient distributions (NOx = nitrate + nitrite; phosphate, $PO_4^{3-}$; silicate, $Si(OH)_4$) in surface waters of Baffin Bay decrease along a west-to-east gradient as they generally follow the ice coverage (Lafond et al., 2019; Tremblay et al., 2002).

Proposed changes:

→ In early summer, macronutrient distributions (NOx = nitrate + nitrite; phosphate, $PO_4^{3-}$; silicate, $Si(OH)_4$) in surface waters of Baffin Bay decrease along a west-to-east gradient as they generally follow the ice coverage (Lafond et al., 2019; Tremblay et al., 2002).

L61: but also dilute due to FW introduction?

Original sentence: The carbonate system of Baffin Bay is influenced by sea ice meltwater, as it has the potential to supply additional alkalinity (AT) (Jones et al., 1983).

Proposed changes:

→ The carbonate system of Baffin Bay is influenced by sea ice meltwater, which has the potential to increase alkalinity (AT) relative to salinity. When the effect of salinity is removed, sea ice meltwater has been shown to supply additional AT in the form of buffering ions such as $[CO_3^{2-}]$ (Jones et al., 1983; Fransson et al., 2023).

L 66: The cited studies do not provide any evidence of impact on shelf or slope nor does Hawkings et al. 2015 look at PP

Original sentence: The export of glacial runoff from the GIS alters shelf and slope waters significantly, thus impacting PP in near-coastal regions (Hawkings et al., 2015; Juul-Pedersen et al., 2015).

Proposed changes:

→ The export of glacial runoff from the GIS alters solute and nutrient delivery to near coastal regions (Hawkings et al., 2015) and the continental shelf (Cape et al., 2019).

L71-72: Two of the cited studies do not provide any data on biological productivity

Original sentence: This fertilizes adjacent marine systems and promotes high levels of biological productivity (Bhatia et al., 2013; Hawkings et al., 2020; Oksman et al., 2022).

Proposed changes:

→ This promotes marine production and a prolonged annual productive season (Oksman et al., 2022).
→ Move Bhatia et al., 2013 to L69

L 109: Were macronutrients filtered?

No, samples for macronutrients were not filtered. The samples were directly frozen after sampling. We will add "without filtration" to the sentence for clarification.

I am not an expert on trace metals but considering the low concentrations expected offshore, how trace-metal clean was the sampling gear?

The Niskin bottles were non-metallic – We will add this information to L104. Comparison with literature values does not suggest any contamination via the sampling gear. LOD and LOQ values also do not suggest contamination via the filtration. The metal-free PFA filtration equipment was acid-washed prior to use as stated in the text.

AOU data is shown but no information is provided on the oxygen calibration?

Two oxygen sensors were part of the CTD system and were calibrated prior to the cruise. By cross-referencing the data of the two sensors, we were able to monitor their drift during the cruise and correct the data accordingly. We will add this information to section 2.2.

Section 3.1. I would show more data in the manuscript instead of tables, potentially transects (not in case there are only 2 stations) or vertical profiles. Now the data is barely presented, or one has to go to supplementary all the time. This does not improve readability.

Please refer to our previous comment. We will include one figure, including ~10 parameters from transect 4. In our opinion, the table holds valuable information, particularly the comparison with literature data.

Generally in discussion, lots of new data and figures are introduced. This should be restructured in my view and moved to results sections. Fe PCA analysis in 4.1.2 and many more figures which follow

Thank you for this recommendation. We had a similar discussion during the writing process and concluded that it is best to present only the "original" data in the results section. This approach provides a general overview for the reader and facilitates future referencing of the data. The PCA analysis was moved to the discussion section because it significantly transforms the data, warranting a detailed discussion. Our aim with Chapter 4.1 was to first introduce the water masses (4.1.1) and then continue with specific characteristics and processes that alter the chemical composition of those water masses (4.1.2). If this aim was not achieved, we will reconsider whether it is reasonable to move Chapter 4.1 to the results section and adjust the discussion accordingly.

L419: July is mid-summer for the arctic

Original sentence: We believe that our sampling near the end of the summer season was crucial for this vast difference in water mass composition.

Proposed changes:

➔ We believe that our sampling during mid-summer season was crucial for this vast difference in water mass composition.

L560: Based on one point in outer part of Disko bay, it is a bit a stretch to make statements on nutrient cycling in Disko bay… In general for section 4.5, very little data is available close to the coast, based on literature it seems many processes are happening inside the fjords, so it seems authors should be very carefull to put those much weight on these few observations concerning the impact of FW runoff

Original sentence: Our results highlight that nutrient cycling in Disko Bay is strongly driven by GIS-derived freshwater input, which stimulates PP and creates an important sink for $CO_2$ long into the summer season.

Proposed changes:

→ Based on the mentioned references and our, albeit, limited data from Disko Bay, we suggest that nutrient cycling in Disko Bay is strongly driven by GIS-derived freshwater input, which stimulates PP and creates an important sink for $CO_2$ long into the summer season.

**Figures & Tables**

Fig 1: Maybe good to integrate figure 1 and 5 in some way and show ice extent (fe with contours)

Thank you for this idea. In our opinion, it is easier for the reader to have these separate figures, as the sea ice chapter is further back in the manuscript. This approach emphasizes the decline of sea ice in the study area over the study period.

Table 1: Silicate values (max) are very high, is that an outlier?

No, the max value (35.95) was found in the water mass BBW, where Sherwood *et al.* (2021) have found similar/even higher silicate values (41.6 ± 25.5; *n* = 31). This holds true for the other macronutrients as well. We will add a sentence about this in the paragraph for clarification.

Fig 2: it would be good to include recent work in West Greenland eg Rysgaard et al 22020

As the classification system for the water masses was largely adapted from Curry et al. (2011), we will include and discuss the updated hydrograph by Rysgaard et al. (2020) for comparison. Additionally, we could include a more detailed TS diagram for the northern and southern regions of the shelf.

Table S1, I assume that water depth are the sampling depths?

We will change this to "Sampling depth [m]".

Fig 4: Instead of plotting vs lat/lon, plot against TS or other chemical values to assess drivers

We will switch the axis in 4 a) and put a compass symbol W-E and in the other diagrams S-N. We will evaluate whether it is reasonable to plot against potential density or salinity, but ultimately, we aim to emphasize the geographical distribution.

---

## Author Comment (AC2)

**Review of Influences on Chemical Distribution Patterns across the West Greenland Shelf: The Roles of Ocean Currents, Sea Ice Melt, and Freshwater Runoff.**

General comment. This work investigates inorganic carbon, macro and micronutrients along the West Greenland Coast, an area facing enhanced changes due to global warming, mainly enhanced glacial melt (both from glaciers and sea ice). It should be highlighted that it is not common to have such a wide spectrum of biogeochemical parameters, and to get insights from such an extensive set of parameters, the authors followed an original approach: PCAs. The sampling strategy is well adapted, as mentioned before, the number of parameters is impressive, the analysis and data treatment are also solid and state of the art, the PCA's approach is original and solid. The authors know the area and what they are doing. However, I have some relatively minor concerns.

We sincerely thank the reviewer for the helpful and constructive comments. These suggestions will significantly improve the revised manuscript, particularly in terms of the presentation of the results. We greatly appreciate the time and effort the reviewer has dedicated to providing this valuable feedback. Thank you for your kind words!

Sometimes, the suggestions or conclusions are not always as strong as the wording suggests, so I would recommend some changes in the text. For example, I found the wording "corrosive", especially in line 563, inappropriate. I understand that it might appear necessary to have some attention-grabbing sentences/ideas, but the point is that if we consider that corrosive waters apply to waters with $\Omega$aragonite or $\Omega$calcite below 1, there are NO corrosive surface waters during that study (Figure S10). If there is no question that the glacial meltwater promotes the decrease of $\Omega$aragonite as it it shown by that study, we cannot draw any conclusion on the rate of the $\Omega$aragonite change, any perspectives of how it will change in the future and when. So I would refrain from this alarming tune. The work is interesting enough without having to oversell the conclusions, I believe.

Original sentence: Apart from the Disko Bay area, the surface water of stations closest to the coast of Greenland were corrosive with lower buffering capacities (refer to Sect. 3.4).

Proposed changes: Apart from the Disko Bay area, the surface water of stations closest to the coast of Greenland had lower $\Omega$ Aragonite values with lower buffering capacities (refer to Sect. 3.4).

➔ Additionally, we will add specific values to line 207. This gives a better comparison between stations 6/11 and station 17 in the following sentence.

Original sentence: The higher CT:AT ratios of coastal stations coincided with higher Revelle factors and lower $\Omega$ Aragonite values, reflecting the lower buffering capacity of coastal waters.

Proposed changes: In surface waters of coastal stations 6 and 11, higher CT:AT ratios (0.920 and 0.925) coincided with higher Revelle factors (12.8 and 13.3) and lower $\Omega$ Aragonite values (1.93 and 1.82), reflecting the lower buffering capacity of coastal waters. In contrast, the surface waters of Disko Bay (station 17; 3 m) had lower CT:AT ratios (0.907), caused by low CT values of 1972 µmol kg$^{-1}$, corresponding to low $p$CO$_2$ values of 255 µatm. Coupled to this, the pH is high (8.19) and the Revelle factor low (11.85), indicating the higher buffering capacity of this water, which is also reflected by a maximum $\Omega$ Aragonite value of 2.17.

There is evidence, well shown in the manuscript, of a higher CT:AT ratio in sea ice meltwater. The impact of ikaite is correctly addressed, and enhanced CT is well mentionned. This high CT is ascribed to enhanced organic carbon remineralisation below sea ice. Why not, but I suggest

mentioning that this excess CT can also come from remineralisation within the ice itself. As it has been correctly mentioned elsewhere, there is an enhanced concentration of trace metals in sea ice. This enhancement comes from both the atmosphere and the scavenging of organic matter in sea ice during its growth. So, it is not only the trace metal concentration that is enhanced in sea ice, but also the organic matter concentration. This comes with enhanced respiration and release of CT within the ice. It could be stated in the manuscript in part 4.4.

Original sentence: Possible explanations for this high surplus in CT could be the release of CT derived from organic carbon remineralization under the ice (Tynan et al., 2016; Bates et al., 2009) and the inflow of additional CT transported through the CAA to Baffin Bay (Shadwick et al., 2011b; Henson et al., 2023; Azetsu-Scott et al., 2010).

Proposed changes: Possible explanations for this high surplus in CT could be the release of CT derived from organic carbon remineralization under the ice (Tynan et al., 2016; Bates et al., 2009) or within the ice as sea ice typically shows higher concentrations in OM than the underlying water (Vancoppenolle et al., 2013). Another source could be the inflow of additional CT transported through the CAA to Baffin Bay (Shadwick et al., 2011b; Henson et al., 2023; Azetsu-Scott et al., 2010).

➔ Additional change to L521

Original sentence: However, the inflow of Pacific-origin waters and remineralization under the ice led to overall higher CT:AT ratios, resulting in higher pCO2 and lower pH and Ω Aragonite values in surface waters.

Proposed changes: However, the inflow of Pacific-origin waters and remineralization under and within the ice led to overall higher CT:AT ratios, resulting in higher pCO2 and lower pH and Ω Aragonite values in surface waters.

Also, despite the very extensive set of parameters, I regret that there are many indirect assumptions about the productivity of the water masses, while a simple measurement of Chl a would have been probably more robust. I presume that if the information is not presented here, it is because it is unavailable. Otherwise, it will be more robust in my mind to show the phytoplanktonic biomass as well, to discuss the productivity of waters, especially at that time of the year.

We currently do not have access to the Chl a data or phytoplanktonic biomass. As this was an interdisciplinary cruise, additional results may be published in the future by other research groups, which could then relate to this publication. Currently, we only have the oxygen and AOU data available.

Minor remarks

Line 169 The calculation was based on the median potential temperature of WGSW (3.23°C) within a depth range of 30 to 50 m, using observed salinity and AT, and the in situ pCO2 as input parameters. Is there any suggestion to chose that value as reference?

A temperature of 3.23°C refers to the median potential temperature of WGSW ($n$ = 465) from the CTD data, filtered for a depth range of 30–50 m. This provides a solid reference value for the temperature of this water mass within this depth range. We will add this information to the text.

Line 235 "The concentration of dFe and dNi in surface waters (Fig. S 9c and e) followed a similar pattern as the surface water nutrient concentrations (Fig. S 7). It is not clear to me to which pattern you are referring. I presume it is the decrease in concentration at the surface. But the

pattern is not very clear to me, at least for dNi. For instance, it is much clearer for dCd, which appears to me to have the clearest "nutrient-like" vertical profile. So, why are you mentioning specifically dNi and not dCd?

Yes, we understand how this is misleading.

Original sentence: The concentration of dFe and dNi in surface waters (Fig. S 9 c and e) followed a similar pattern as the surface water nutrient concentrations (Fig. S 7). The concentrations were higher close to the coast, decreased with distance along each transect and increased again towards the west of each transect. The surface water concentrations of dCd were rather uniform and increased only towards the west of each transect (Fig. S 9 g).

Proposed changes: The concentration of dFe and dNi in surface waters (Fig. S 9 c and e) were higher close to the coast, decreased with distance along each transect and increased again towards the west of each transect. The surface water concentrations of dCd were rather uniform across the shelf and increased only towards the west of each transect (Fig. S 9 g).

→ There is a difference in the coastal stations. Coastal freshwater runoff was not a source of dCd, while providing dFe and dNi.

3.4 Carbonate system parameters. Line 255 to 280. I have no concern about your general description of results, but my attention has been caught by the shift in CT:AT at the surface waters that might be addressed, especially station 23 (or 24), 14 and 05, with related impact on Revelle factor and aragonite saturation. I am wondering if you can address this striking/questioning feature.

We think this is referring to L275:

"The westernmost surface waters of each transect showed lower AT and CT values with overall highest CT:AT ratios in surface waters. Consequently, pCO2 values and Revelle factors followed this trend and were higher at these stations, whereas pH and Ω Aragonite values were lower."

We addressed this in chapter 4.4 from L500 onwards by calculating salinity-normalized AT and temperature-salinity-normalized CT. We found that even though sea ice meltwater is diluting AT and CT, it provides a small source of AT and CT to the surface waters (the sea ice endmember is not 0). Overall, the CT:AT ratios are higher, so there must be an additional source of CT, which we concluded to be remineralization under the ice* and the inflow of Pacific-origin waters with high CT values.

We concluded this in the following statement (L520):

"Our results show that the dissolution of ikaite in sea ice acted as a source of AT, providing a small geochemical buffer to meltwater-influenced surface waters (Jones et al., 2021). However, the inflow of Pacific-origin waters and remineralization under the ice* led to overall higher CT:AT ratios, resulting in higher pCO2 and lower pH and Ω Aragonite values in surface waters."

*see proposed changes above: and within the ice

Line 329 "and sea ice 330 meltwater as a dCu source". While the source of Cu from glacial freshwater seems compelling from Figure S12, this is not the case for meltwater sea ice, as the value of dCu is the same at salinity 33 and 30.5-31. I acknowledge that we can expect sea ice meltwater from sea ice, according to current understanding of trace metals in Antarctic sea ice and Arctic sea ice (Tovar-Sánchez et al., 2010). But this does not seem obvious to me in the current survey, at least from the figures S12, despite the PCAs.

Yes, this is a very good point. The influence of sea ice is evident when examining the surface water concentrations, particularly at Stations 24 and 30 (Fig. S9f) compared to the adjacent stations. The surface water concentrations of dCu are higher towards the west of Transects 4 and 5. Specifically, at Station 24, the concentrations are 206 ng/L at 3 m and 204 ng/L at 36 m, compared to Station 22, which has 157 ng/L at 6 m. Similarly, at Station 30, the concentration is 185 ng/L at 3 m, whereas at Station 28, it is 161 ng/L at 3 m.

Original sentence: The distribution of dCu correlated negatively with salinity and was influenced by coastal freshwater input and sea ice meltwater as a dCu source (Fig. S 12 d).

Proposed changes: The distribution of dCu correlated negatively with salinity and was mainly influenced by coastal freshwater input (Fig. S12d). Similarly, sea ice meltwater can be identified as a dCu source to surface waters (Fig. S9f), with dCu concentrations increasing towards the west of Transects 4 and 5.

Figure 4 a). I found it confusing to plot longitude from low to high, as this implies that the West is on the left and the East is on the right, especially as there is a map in the same figure, meaning that the East is on the left in fig 4e) and on the right in fig 4a.

We will switch the axis in 4 a) and put a compass symbol W-E and in the other diagrams S-N to improve readability and logic of the figure.

Line 410. How robust is the comparison of AOU values in WGSW and AW? It is not clear to me if you are comparing similar depths. I believe this is not the case, and you are comparing the whole data set without considering the depth. I am not comfortable with that, as depth is one of the first predictors of AOU. In addition, the AOU is also affected by air-sea exchange, which depends on ice cover. So, in my opinion, the suggestion that warmer shelf waters being more productive than the colder AW, is not well supported by the comparison of AOU, even if I expect that is indeed correct.

Yes, this is completely true. We addressed a similar point by focusing on a depth range of 30-50 m for NOx and CT. Please see below for the proposed changes, including some rearrangement of the following sentences. However, we are still discussing this point and might remove this argument.

Original sentence: Lower AOU values in WGSW (-4 µmol L-1) than in AW (36 µmol L-1) indicate that the warmer shelf waters are more productive than the colder AW.

Proposed changes: As AOU is a depth-dependent variable, we propose examining a depth range of 30 to 50 m for a better comparison between WGSW and AW. The median AOU values within this range are -6 $\mu mol\ L^{-1}$ in WGSW and 7 $\mu mol\ L^{-1}$ in AW. This indicates that the warmer shelf waters are more productive than the colder AW. As primary production was not measured directly in this study, we support this argument with a study by Krawczyk et al. (2021), that found the highest values of chlorophyll *a* along the south-west Greenland coast and the lowest values along Baffin Island. This is also in agreement with overall lower nutrient concentrations in WGSW than AW (refer to Sect. 4.1.2 and Table 2), as nutrients are removed from the water column through PP. Fig. 4 a shows the NOx concentrations against longitude in a depth range of 30 to 50 m, including literature data from the Green Edge cruise in 2016 (Bruyant et al., 2022). There is a clear difference between lower NOx concentrations in WGSW on the shelf and higher NOx concentrations in AW further offshore in areas that were still ice-covered. This gradient in nutrient concentrations is caused by the direction of the sea ice retreat from east to west and has been linked to higher productivity and species richness along the southern coast of west

Greenland and in Disko Bay, compared to Baffin Island and northwest Greenland (Krawczyk et al., 2021; Lafond et al., 2019). We believe that our sampling near the end of the summer season was crucial for this vast difference in water mass composition. Shelf waters had already become depleted in nutrients, while further offshore biological processes only increased recently because of the retreating sea ice (refer to Sect. 4.4). With depth, respiration returns nutrients to the water column as seen by high AOU values in WGIW (62 µmol L-1) and overall higher nutrient concentrations (refer to Table 2).

Line 597: "Thus, the benthic ecosystem of the west Greenland shelf is potentially vulnerable to future ocean acidification and suppression of CaCO3 saturation states." There is a bit of mixing, and confusion introduced there, in my opinion. There is only on single data below the aragonite saturation at 700m. This does not correspond to the condition on the shelf. What means "suppression of CaCO3 saturation states". Is it possible to "suppress" any saturation state? I am not a native English speaker, but the expression is a bit odd to me. I would have expected something like undersaturation in Aragonite, decrease in saturation state, but not a suppression.

The information about corrosive conditions and Ω Aragonite undersaturation is already given in L596, so we are changing the sentence.

Original sentence: Thus, the benthic ecosystem of the west Greenland shelf is potentially vulnerable to future ocean acidification and suppression of $CaCO_3$ saturation states.

Proposed changes: Thus, the benthic ecosystem of the west Greenland shelf might be negatively impacted, particularly becoming unfavorable for shell-forming marine organisms.

Line 604: I appreciated the part "We hypothesize that this return flux decreased with increasing distance 605 from the Greenlandic coast as the amount of exported biogenic and terrigenous material available for remineralization and reversible scavenging also decreased. Further studies looking into Baffin Bay sediments are needed to disclose benthic fluxes of trace elements and reactions occurring therein. Our study provides a first insight into the importance of benthic inputs for trace element cycling on the west Greenland shelf and Baffin Bay." The evidence of the process, as derived from the PCAs is not fully compelling. I am not against the idea, but at that point, I appreciate that it is mentioned as only a suggestion.

This is only an idea so far but since a similar process was found in the East Sea, we thought this could be a hypothesis. More information about this is given in L384: "A similar observation was made for dMn, dFe, and dCo in the Ulleung Basin of the East Sea by Seo et al. (2022). Concentrations were significantly higher on the slopes and bottom layer of the Ulleung Basin in contrast to the Japan Basin, which the authors associated with a large sedimentary release by OM degradation caused by the high OM content of the shelf sediments (Seo et al., 2022)."

Figure S3. Something is wrong with the plots. A large white square masks a large part of them. I don't know where the depth scale of the plot is.

These are water column profiles of Transect 3. Station 15 is displayed separately because it is further south than the other stations. Although the depth scale is the same on both graphs, we recognize that this presentation can be confusing. We will remodel the plots to make this clearer.

I found the Table S5 odd, with values missing (max salinity station 15 to 30); Pot. Temp minimum station 8 to 17, and so on. In addition, I would have expected a single value as minimum or maximum, rather than a range of values. Per se, a minimum value is only one value. The values are given for only some stations and not all. For example, there are no salinity values for 6 to 10,

but there is a maximum potential temperature value for that station, but not a minimum value. It looks a bit random. Also, I'm wondering if the table would benefit from being rotated 90°. I mean, one line per station, and then each column corresponding to a parameter (min salinity, maximum salinity...., etc.).

For this table, we examined the overall minimum and maximum values across the entire data set, specifically the 1% lowest and highest values overall. However, we understand how this presentation can be misleading. We will adapt your recommendations and create a table with the minimum and maximum values for each station.

Figure S1. This plot is made with only 2 stations parallel to the coast, while all other plots are based on transects of at least 5 stations perpendicular to the coast. I don't think that it is very useful to have it. And I'm sure the structures exhibited by the contour lines for salinity and oxygen have no physical meaning and can be removed.

We will change these plots to vertical profiles with two lines for St. 1 and 2 (similar to Fig. S 6).

Figure S9, S10 and others. I won't cut the symbol above the 0 depth. It makes the surface a little less easy to read. It looks like you applied a "blank" patch above the 0-depth line.

We will make the space above 0 m bigger.

---

## Author Response (AR1)

**Response to reviewer I**

The paper investigates the key drivers of chemical distribution patterns on the West Greenland shelf during summer. The study focuses on how ocean currents, sea ice melt, and terrestrial freshwater runoff affect the distribution of macronutrients, carbonate system parameters, and dissolved trace elements based on a cruise with CTD casts, water sampling for nutrients and carbonate chemistry, and trace element analysis. The study area spans across several transects along the Greenland shelf.

We sincerely thank the reviewer for the helpful and constructive comments. These suggestions have significantly improved the revised manuscript, particularly in terms of its overall structure and the presentation of the results. We greatly appreciate the time and effort the reviewer has dedicated to providing this valuable feedback.

While the research aims to provide valuable insights into this complex marine environment, the presentation of the study suffers from significant shortcomings in readability, structure, and data visualization. Sentences are often long, lacking a clear flow of ideas jumping between concepts without sufficient transition or explanation, especially in the result and discussion section.

Thank you for this feedback. We significantly improved the wording of the text and included an additional figure to the results section (see comment below) to improve the visualization. This figure shows section plots of selected parameters from transect 4, giving the reader a better understanding of the results.

Personally I feel the data is not presented effectively. While there are figures in the supplementary material, the main text lacks visuals to illustrate the actual data. Instead, it overuses tables, which don't show the data as clearly as figures would. Also a lot of new data and figures are presented in the discussion which makes readability harder.

In the results section, we included a figure with multiple panels displaying various parameters (e.g. Sal, Temp, Oxygen, NOx, AT, CT, dV, dMn, and dFe) for Transect 4. This figure helps visualize the data and introduce general concepts of the study area, highlighting similarities between the parameters (e.g. NOx and dCd, dMn and dCo).

Additionally, the referencing throughout the manuscript needs improvement. Several citations are inappropriate, not directly supporting the statements they accompany (See below as well). Key references relevant to the specific processes and region under study are also missing. For instance, the manuscript would benefit from including more recent and relevant work on the hydrography of the West Greenland shelf by Rysgaard et al. (2020)

We corrected the citation in the introduction as seen below. We included the hydrography work by Rysgaard et al. (2020) in chapter 3.5 by comparing this updated view with the categorization system used throughout the rest of the manuscript. The reason for using the older system was that it gives clear boundaries for the water masses that we used to describe and compare the chemical signatures throughout the discussion.

**Below some specific comments**

L20: sea ice retreat creates nutrient gradients: in general you have to be aware that would you present is just a snapshot of july, so be careful to make statements like this. Yes, there is a gradient in July, but not necessarily for August, September,..

**Changes:**

- → L15 "during late summer" change to "during July".
- → During the study period, we were able to capture a distinct nutrient gradient following the east-to-west direction of sea ice retreat, with low nutrient levels in highly productive shelf waters and high nutrient levels in areas with prolonged ice cover.

L 33: This is not supported by paper you cite here AND L34-35: Look at newer circulation paper in West Greenland as well, eg Rysgaard et al. 2020

**Changes:**

→ The circulation across the west Greenland shelf and Baffin Bay is dominated by two major currents in the region: the southward-moving Baffin Island Current (BIC) along the Baffin Island continental slope and the northward-moving West Greenland Current (WGC) along the west Greenland continental slope (Rysgaard et al., 2020).

**L47: suggest to look at more appropriate referencing for sea-ice conditions**

Krawczyk et al., 2021 use satellite-derived sea ice concentration as monthly average from 1978–2014, which is an appropriate reference. We also included a reference that has a more global perspective and one that looks at sea ice in Baffin Bay in October and the changes of the melt period.

**Changes:**

→ In the last two decades, sea ice concentrations in Baffin Bay and the Labrador Sea have declined significantly due to climate change (Krawczyk et al., 2021), additionally driven by an earlier melt onset and delayed freeze-up (Stroeve and Notz, 2018; Ballinger et al., 2022).

**L49: Do the studies provide evidence of this?**

Yes, both do.

**Changes:**

→ The reduction in sea ice cover, combined with increased inflow of Atlantic-sourced waters (Krawczyk et al., 2021) and elevated freshwater discharge (Møller et al., 2023), has been associated with an overall increase in PP and species richness along the west Greenland shelf (Krawczyk et al., 2021; Møller et al., 2023).

**L55: in early-summer**

**Changes:**

→ In early summer, macronutrient distributions (NOx = nitrate + nitrite; phosphate, PO43-; silicate, Si(OH)4) in surface waters of Baffin Bay decrease along a west-to-east gradient as they generally follow the ice coverage (Lafond et al., 2019; Tremblay et al., 2002).

**L61: but also dilute due to FW introduction?**

**Changes:**

→ Sea ice meltwater also influences the carbonate system of Baffin Bay by increasing alkalinity (AT) relative to salinity. When the effect of salinity is removed, sea ice

meltwater has been shown to supply additional AT in the form of buffering ions such as carbonate (CO32–) (Jones et al., 1983; Fransson et al., 2023).

L 66: The cited studies do not provide any evidence of impact on shelf or slope nor does Hawkings et al. 2015 look at PP

**Changes:**

→ The export of glacial runoff from the GIS modifies the delivery of solutes and nutrients to near coastal regions (Hawkings et al., 2015) and the continental shelf (Cape et al., 2019). This glacier-derived freshwater alters marine concentrations of macronutrients (Hawkings et al., 2015; Hawkings et al., 2016; Hawkings et al., 2017; Hendry et al., 2019) and dissolved trace elements (Bhatia et al., 2013; Hawkings et al., 2020; Krause et al., 2021), influences local hydrography, and promotes the upwelling of nutrient-rich deep waters (Meire et al., 2017; Juul-Pedersen et al., 2015).

L71-72: Two of the cited studies do not provide any data on biological productivity

**Changes:**

→ These processes enhance marine productivity and extend the duration of the annual productive season (Oksman et al., 2022).

**L 109: Were macronutrients filtered?**

No, samples for macronutrients were not filtered. The samples were directly frozen after sampling. We added "without filtration" to the sentence for clarification.

I am not an expert on trace metals but considering the low concentrations expected offshore, how trace-metal clean was the sampling gear?

The Niskin bottles were non-metallic – We added this information to L113. Comparison with literature values does not suggest any contamination via the sampling gear. LOD and LOQ values also do not suggest contamination via the filtration. The metal-free PFA filtration equipment was acid-washed prior to use as stated in the text.

**AOU data is shown but no information is provided on the oxygen calibration?**

Two oxygen sensors were part of the CTD system and calibrated prior to the cruise. By cross-referencing the data of the two sensors, we were able to monitor their drift during the cruise and correct the data accordingly. We added this information to section 2.2.

Section 3.1. I would show more data in the manuscript instead of tables, potentially transects (not in case there are only 2 stations) or vertical profiles. Now the data is barely presented, or one has to go to supplementary all the time. This does not improve readability.

Please refer to our previous comment. We included one figure, including 9 parameters from transect 4 to the results section.

Generally in discussion, lots of new data and figures are introduced. This should be restructured in my view and moved to results sections. Fe PCA analysis in 4.1.2 and many more figures which follow

Thank you for this recommendation. We had a similar discussion during the writing process and concluded that it is best to present only the "original" data in the results section. This approach provides a general overview for the reader and facilitates future referencing of the data. The PCA

analysis is part of the discussion section because it significantly transforms the data, warranting a detailed discussion. However, we moved the section about water masses to the results section to make the discussion clearer.

**L419: July is mid-summer for the arctic**

**Changes:**

→ We consider the timing of our sampling during the mid-summer season to be critical in capturing these pronounced differences in water mass composition.

L560: Based on one point in outer part of Disko bay, it is a bit a stretch to make statements on nutrient cycling in Disko bay... In general for section 4.5, very little data is available close to the coast, based on literature it seems many processes are happening inside the fjords, so it seems authors should be very carefull to put those much weight on these few observations concerning the impact of FW runoff

Besides our own data, we base this conclusion on two other studies from Disko Bay and one from Godthåbsfjord (L595 onwards).

**Changes:**

→ Based on these findings and supporting literature, we propose that nutrient cycling in Disko Bay is strongly influenced by GIS-derived freshwater input. This input stimulates PP and establishes Disko Bay as an important seasonal sink for atmospheric CO₂ well into the summer.

**Figures & Tables**

Fig 1: Maybe good to integrate figure 1 and 5 in some way and show ice extent (fe with contours)

Thank you for this idea. In our opinion, it is easier for the reader to have these separate figures, as the sea ice chapter is further back in the manuscript. This approach emphasizes the decline of sea ice in the study area over the study period.

**Table 1: Silicate values (max) are very high, is that an outlier?**

No, the max value (35.95) was found in the water mass BBW, where Sherwood *et al.* (2021) have found similar/even higher silicate values (41.6  $\pm$  25.5; n = 31). This holds true for the other macronutrients as well. added a sentence about this in the paragraph for clarification.

**Changes:**

→ Below the biological productive zone, nutrient concentrations gradually increased with depth, reaching maximum concentrations at station 15 (667 m) (refer to Table 1). Sherwood et al. (2021) reported similarly high nutrient concentrations in these deep waters, which are further discussed in Sect. 4.1.2.

**Fig 2: it would be good to include recent work in West Greenland eg Rysgaard et al 22020**

As the classification system for the water masses was largely adapted from Curry et al. (2011), we included and discussed the updated hydrograph by Rysgaard et al. (2020) for comparison.

**Table S1, I assume that water depth are the sampling depths?**

Yes, thank you. We changed this to "Sampling depth [m]".

Fig 4: Instead of plotting vs lat/lon, plot against TS or other chemical values to assess drivers

We switched the axis in 4 a) and put a compass symbol W-E and in the other diagrams S-N. We plotted 4 c) and d) against salinity and indicated the difference in CT median concentrations with an arrow.

**Response to reviewer II**

General comment. This work investigates inorganic carbon, macro and micronutrients along the West Greenland Coast, an area facing enhanced changes due to global warming, mainly enhanced glacial melt (both from glaciers and sea ice). It should be highlighted that it is not common to have such a wide spectrum of biogeochemical parameters, and to get insights from such an extensive set of parameters, the authors followed an original approach: PCAs. The sampling strategy is well adapted, as mentioned before, the number of parameters is impressive, the analysis and data treatment are also solid and state of the art, the PCA's approach is original and solid. The authors know the area and what they are doing. However, I have some relatively minor concerns.

We sincerely thank the reviewer for the helpful and constructive comments. These suggestions have significantly improved the revised manuscript, particularly in terms of the presentation of the results. We greatly appreciate the time and effort the reviewer has dedicated to providing this valuable feedback. Thank you for your kind words!

Sometimes, the suggestions or conclusions are not always as strong as the wording suggests, so I would recommend some changes in the text. For example, I found the wording "corrosive", especially in line 563, inappropriate. I understand that it might appear necessary to have some attention-grabbing sentences/ideas, but the point is that if we consider that corrosive waters apply to waters with  $\Omega$  aragonite or  $\Omega$  calcite below 1, there are NO corrosive surface waters during that study (Figure S10). If there is no question that the glacial meltwater promotes the decrease of  $\Omega$  aragonite as it it shown by that study, we cannot draw any conclusion on the rate of the  $\Omega$  aragonite change, any perspectives of how it will change in the future and when. So I would refrain from this alarming tune. The work is interesting enough without having to oversell the conclusions, I believe.

Changes: Apart from the Disko Bay area, surface waters at stations closest to the Greenland coast exhibited lower  $\Omega$  Aragonite values with lower buffering capacities (refer to Sect. 3.4).

→ Additionally, we added specific values to line 207. This gives a better comparison between stations 6/11 and station 17 in the following sentence.

Changes: In coastal surface waters at stations 6 and 11, higher CT:AT ratios (0.920 and 0.925) coincided with higher Revelle factors (12.8 and 13.3) and lower  $\Omega$  Aragonite values (1.93 and 1.82), indicating a reduced buffering capacity in coastal waters. In contrast, surface waters in Disko Bay (station 17; 3 m) exhibited a lower CT:AT ratio (0.907), driven by a low CT concentration of 1972 μmol kg-1 and a corresponding low pCO2 value of 255 μatm. This was accompanied by a high pH (8.19), a low Revelle factor (11.85), and a maximum  $\Omega$  Aragonite value of 2.17, indicating the higher buffering capacity of this water.

There is evidence, well shown in the manuscript, of a higher CT:AT ratio in sea ice meltwater. The impact of ikaite is correctly addressed, and enhanced CT is well mentionned. This high CT is ascribed to enhanced organic carbon remineralisation below sea ice. Why not, but I suggest mentioning that this excess CT can also come from remineralisation within the ice itself. As it has been correctly mentioned elsewhere, there is an enhanced concentration of trace metals in sea ice. This enhancement comes from both the atmosphere and the scavenging of organic matter in sea ice during its growth. So, it is not only the trace metal concentration that is enhanced in sea ice, but also the organic matter concentration. This comes with enhanced respiration and release of CT within the ice. It could be stated in the manuscript in part 4.4.

Changes: Possible explanations for the observed surplus in CT include the release of CT from organic carbon remineralization under the ice (Tynan et al., 2016; Bates et al., 2009), or within the ice itself, as sea ice is known to contain higher concentrations in OM than the underlying seawater (Vancoppenolle et al., 2013). Additionally, the inflow of CT-enriched waters transported through the CAA into Baffin Bay may contribute to the elevated CT levels (Shadwick et al., 2011b; Henson et al., 2023; Azetsu-Scott et al., 2010).

**→ Additional change to L521**

Changes: However, the inflow of Pacific-origin waters and remineralization under and within the ice contributed to elevated CT:AT ratios, resulting in higher pCO2 and lower pH and  $\Omega$  Aragonite values in surface waters.

Also, despite the very extensive set of parameters, I regret that there are many indirect assumptions about the productivity of the water masses, while a simple measurement of Chl a would have been probably more robust. I presume that if the information is not presented here, it is because it is unavailable. Otherwise, it will be more robust in my mind to show the phytoplanktonic biomass as well, to discuss the productivity of waters, especially at that time of the year.

We currently do not have access to the Chl a data or phytoplanktonic biomass. As this was an interdisciplinary cruise, additional results may be published in the future by other research groups, which could then relate to this publication. Currently, we only have the oxygen and AOU data available.

**Minor remarks**

Line 169 The calculation was based on the median potential temperature of WGSW (3.23°C) within a depth range of 30 to 50 m, using observed salinity and AT, and the in situ pCO2 as input parameters. Is there any suggestion to chose that value as reference?

A temperature of 3.23°C refers to the median potential temperature of WGSW (n = 465) from the CTD data, filtered for a depth range of 30–50 m. This provides a solid reference value for the temperature of this water mass within this depth range. We added this information to the text (L179).

Line 235 "The concentration of dFe and dNi in surface waters (Fig. S 9c and e) followed a similar pattern as the surface water nutrient concentrations (Fig. S 7). It is not clear to me to which pattern you are referring. I presume it is the decrease in concentration at the surface. But the pattern is not very clear to me, at least for dNi. For instance, it is much clearer for dCd, which appears to me to have the clearest "nutrient-like" vertical profile. So, why are you mentioning specifically dNi and not dCd?

Yes, we understand how this is misleading.

Changes: Surface concentrations of dFe and dNi (Fig. S 9 c and e) were elevated near the coast, decreased with increasing distance along each transect and increased again towards the west of each transect. In contrast, surface dCd concentrations were rather uniform across the shelf, with a slight increase toward the westernmost stations (Fig. S 9 g).

→ There is a difference in the coastal stations. Coastal freshwater runoff was not a source of dCd, while providing dFe and dNi.

3.4 Carbonate system parameters. Line 255 to 280. I have no concern about your general description of results, but my attention has been caught by the shift in CT:AT at the surface waters that might be addressed, especially station 23 (or 24), 14 and 05, with related impact on Revelle factor and aragonite saturation. I am wondering if you can address this striking/questioning feature.

We think this is referring to L275:

"In the westernmost surface waters of each transect, AT and CT values were again lower, but CT:AT ratios were among the highest observed. As a result, pCO2 values and Revelle factors were elevated at these stations, while pH and  $\Omega$  Aragonite values were reduced."

We addressed this in chapter 4.3 from L543 onwards by calculating salinity-normalized AT and temperature-salinity-normalized CT. We found that even though sea ice meltwater is diluting AT and CT, it provides a small source of AT and CT to the surface waters (the sea ice endmember is not 0). Overall, the CT:AT ratios are higher, so there must be an additional source of CT, which we concluded to be remineralization under and within the ice and the inflow of Pacific-origin waters with high CT values.

We concluded this in the following statement (L564):

"Our results indicate that the dissolution of ikaite in sea ice acted as a source of AT, providing a small geochemical buffer to surface waters (Jones et al., 2021). However, the inflow of Pacificorigin waters and remineralization under and within the ice contributed to elevated CT:AT ratios, resulting in higher pCO2 and lower pH and  $\Omega$  Aragonite values in surface waters."

Line 329 "and sea ice 330 meltwater as a dCu source". While the source of Cu from glacial freshwater seems compelling from Figure S12, this is not the case for meltwater sea ice, as the value of dCu is the same at salinity 33 and 30.5-31. I acknowledge that we can expect sea ice meltwater from sea ice, according to current understanding of trace metals in Antarctic sea ice and Arctic sea ice (Tovar-Sánchez et al., 2010). But this does not seem obvious to me in the current survey, at least from the figures S12, despite the PCAs.

Yes, this is a very good point. The influence of sea ice is evident when examining the surface water concentrations, particularly at Stations 24 and 30 (Fig. S9f) compared to the adjacent stations. The surface water concentrations of dCu are higher towards the west of Transects 4 and 5. Specifically, at Station 24, the concentrations are 206 ng/L at 3 m and 204 ng/L at 36 m, compared to Station 22, which has 157 ng/L at 6 m. Similarly, at Station 30, the concentration is 185 ng/L at 3 m, whereas at Station 28, it is 161 ng/L at 3 m.

Changes: The distribution of dCu correlated negatively with salinity and was primarily influenced by coastal freshwater input (Fig. S 12d). Sea ice meltwater also contributed to elevated dCu concentrations in surface waters (Fig. S 9 f), particularly toward the western ends of Transects 4 and 5.

Figure 4 a). I found it confusing to plot longitude from low to high, as this implies that the West is on the left and the East is on the right, especially as there is a map in the same figure, meaning that the East is on the left in fig 4e) and on the right in fig 4a.

We switched the axis in 4 a) and put a compass symbol W-E and in the other diagrams S-N. We plotted 4 c) and d) against salinity and indicated the difference in CT median concentrations with an arrow.

Line 410. How robust is the comparison of AOU values in WGSW and AW? It is not clear to me if you are comparing similar depths. I believe this is not the case, and you are comparing the whole data set without considering the depth. I am not comfortable with that, as depth is one of the first predictors of AOU. In addition, the AOU is also affected by air-sea exchange, which depends on ice cover. So, in my opinion, the suggestion that warmer shelf waters being more productive than the colder AW, is not well supported by the comparison of AOU, even if I expect that is indeed correct.

Yes, this is completely true. We addressed a similar point by focusing on a depth range of 30-50 m for NOx and CT. Please see below for the proposed changes, including some rearrangement of the following sentences.

Changes: As AOU is a depth-dependent variable, we propose examining a consistent depth range of 30 to 50 m to enable a more meaningful comparison between WGSW and AW. Within this range, the median AOU values are -6  $\mu$ mol L-1 for WGSW and 7  $\mu$ mol L-1 for AW. This suggests that warmer shelf waters (WGSW) are more biologically productive than colder AW, as indicated by greater oxygen production in the upper water column. Although primary production was not directly measured in this study, this interpretation is supported by Krawczyk et al. (2021), who reported the highest chlorophyll a concentrations along the southwest Greenland coast and the lowest values along Baffin Island. This pattern is consistent with overall lower nutrient concentrations observed in WGSW compared to AW (refer to Sect. 4.1.2 and Table 2), as nutrients are removed from the water column during PP. Fig. 5 represents NOx concentrations plotted against longitude within the 30 to 50 m depth range, including literature data from the Green Edge cruise in 2016 (Bruyant et al., 2022). A clear spatial gradient is evident, with lower NOx concentrations in WGSW on the shelf and higher NOx concentrations in AW further offshore, particularly in areas that were still ice-covered. This nutrient gradient reflects the eastto-west progression of sea ice retreat and has been linked to enhanced PP and species richness along the southern coast of west Greenland and in Disko Bay, compared to the more nutrientrich but less productive regions near Baffin Island and northwest Greenland (Krawczyk et al., 2021; Lafond et al., 2019). We consider the timing of our sampling during the mid-summer season to be critical in capturing these pronounced differences in water mass composition. By this time, shelf waters had already become depleted in nutrients, whereas offshore regions were only beginning to experience increased biological processes following the retreat of sea ice (refer to Sect. 4.3). With increasing depth, respiration returns nutrients to the water column, as evidenced by high AOU values in WGIW (62 µmol L-1) and overall higher nutrient concentrations at depth (refer to Table 2).

Line 597: "Thus, the benthic ecosystem of the west Greenland shelf is potentially vulnerable to future ocean acidification and suppression of CaCO3 saturation states." There is a bit of mixing, and confusion introduced there, in my opinion. There is only on single data below the aragonite saturation at 700m. This does not correspond to the condition on the shelf. What means "suppression of CaCO3 saturation states". Is it possible to "suppress" any saturation state? I am not a native English speaker, but the expression is a bit odd to me. I would have expected something like undersaturation in Aragonite, decrease in saturation state, but not a suppression.

Changes: This accumulation drives corrosive conditions, resulting in  $\Omega$  Aragonite undersaturation in deep waters of southern Baffin Bay. Such conditions pose a potential threat to the benthic ecosystem, particularly by creating an unfavorable environment for shell-forming marine organisms.

Line 604: I appreciated the part "We hypothesize that this return flux decreased with increasing distance 605 from the Greenlandic coast as the amount of exported biogenic and terrigenous material available for remineralization and reversible scavenging also decreased. Further studies looking into Baffin Bay sediments are needed to disclose benthic fluxes of trace elements and reactions occurring therein. Our study provides a first insight into the importance of benthic inputs for trace element cycling on the west Greenland shelf and Baffin Bay." The evidence of the process, as derived from the PCAs is not fully compelling. I am not against the idea, but at that point, I appreciate that it is mentioned as only a suggestion.

This is only an idea so far but since a similar process was found in the East Sea, we thought this could be a hypothesis. More information about this is given in L384: "A similar pattern was observed by Seo et al. (2022) in the East Sea, where significantly higher concentrations of dMn, dFe, and dCo were found on the slopes and in the bottom layer of the Ulleung Basin compared to the Japan Basin. The authors attributed this to enhanced sedimentary release driven by the high OM content of shelf sediments (Seo et al., 2022)."

Figure S3. Something is wrong with the plots. A large white square masks a large part of them. I don't know where the depth scale of the plot is.

We remodeled the plots to make this clearer. Station 15 is displayed separately because it is further south than the other stations. The depth scale is now shown for both graphs.

I found the Table S5 odd, with values missing (max salinity station 15 to 30); Pot. Temp minimum station 8 to 17, and so on. In addition, I would have expected a single value as minimum or maximum, rather than a range of values. Per se, a minimum value is only one value. The values are given for only some stations and not all. For example, there are no salinity values for 6 to 10, but there is a maximum potential temperature value for that station, but not a minimum value. It looks a bit random. Also, I'm wondering if the table would benefit from being rotated 90°. I mean, one line per station, and then each column corresponding to a parameter (min salinity, maximum salinity,..., etc.).

We simplified this table and reduced the number of minimum and maximum values.

Figure S1. This plot is made with only 2 stations parallel to the coast, while all other plots are based on transects of at least 5 stations perpendicular to the coast. I don't think that it is very useful to have it. And I'm sure the structures exhibited by the contour lines for salinity and oxygen have no physical meaning and can be removed.

We changed these plots to vertical profiles with two lines for St. 1 and 2 (similar to Fig. S 6).

Figure S9, S10 and others. I won't cut the symbol above the 0 depth. It makes the surface a little less easy to read. It looks like you applied a "blank" patch above the 0-depth line.

We included more space above 0 m for all plots.

---

## Author Response (AR2)

We would like to sincerely thank the reviewer for their thoughtful and constructive feedback on our manuscript. We greatly appreciate the time and effort dedicated to evaluating our work. Below, we address each point raised and describe the corresponding revisions made.

1) I follow the comment by one of the referees that many of the figures only appear in the discussion. I think the figures in the discussion are appropriate, but I am missing the figures in the results section, while the results figures are added to the supplement. While I agree that the surface figures (S7, S9 and S11) are best placed in a supplement (the information discussed in the text can be drawn from the profiles together with Figure 1), I would suggest to move Figures S6, S8 and S10 to the main text, to make it easier for the reader to follow the results section, without continuously having to go back and forth between main text and supplement.

We moved the figures to the appropriate sections.

2) I also think that Figures S1-S5 can be combined into one or two figure(s) showing temperature, salinity and oxygen (row) for each transect (column). I explicitly left out potential density here as this is not discussed in the main text (section 3) and can thus remain in the supplement. This new suggested figure in the main body could then replace Figure 2.

We combined the water profiles as suggested and included them as Fig. 2.

3) Line 69: first appearance of pCO2 - please replace with partial pressure of CO2 (pCO2)

This was changed as suggested.

4) Line 172: "Limits of detection (LOD) and limits of quantification (LOQ)" - please briefly explain what this means

We added the following definition (L174): "The LOD is defined as three times the standard deviation ( $3 \times SD$ ) of the blank measurements and represents the lowest concentration that can be reliably distinguished from background noise, though not necessarily quantified with precision. The LOQ is defined as ten times the standard deviation ( $10 \times SD$ ) and represents the lowest concentration that can be quantitatively determined with acceptable accuracy and precision (DIN e.V., 2008)."

5) Line 185: "Values below LOD were replaced with a random value between zero 185 and LOD." I think this needs some more explanation/justification (e.g. in form of a reference) for the wider audience

We added two references about methods for replacing data below LOD/LOQ (L188): "As a minor proportion of the total observations (< 3 %) were below the LOD or LOQ, a single imputation method was applied (Helsel, 2005; Jain, 2016). Values below the LOD were replaced with a random value between zero and the LOD, while values below the LOQ were replaced with a random value between the LOD and LOO."

6) Line 189: "Guttman-Kaiser" criterion and "broken stick analysis" - please explain in more depth or add a reference for the wider audience

We added the appropriate references (L193): "Three principal components (PCs) were selected based on the Guttman-Kaiser criterion and the scree plot, retaining only those with eigenvalues

greater than 1 (Bro and Smilde, 2014). A broken stick analysis was performed to evaluate the significance of variable loadings and identify which variables are associated with specific PCs (Peres-Neto et al., 2003)."

---

## Author Response (AR3)

**Comment by referee:**

Line 71. « However, on an annual cycle, sea ice does not serve as a net source or sink of these species to the underlying seawater. Instead, it serves as a mechanism for temporal and spatial redistribution, as these species are trapped during ice formation in autumn and winter, and subsequently released during ice melt – potentially at a different location due to drifting sea ice (Thomas et al., 2011) ".

This sentence was added during the review process. I'm not sure why. I understand the overall idea, but it is too simple and it ignores the transport of DIC and TA at depth during the brine expulsion and related deep water formation (Rysgaard et al. 2011; Crabeck O. et al. 2025; Moreau et al. 2016; Grimm et al. 2016), the fluxes of CO2 to the atmosphere that alter the DIC content (Geilfus et al. 2014; Nomura et al. 2013; 2014; 2018), or even sea ice remineralisation (Zhou et al. 2015) that transforms POC/DOC into DIC. I would remove that new sentence.

Thank you for the helpful comment. We agree that examining different timescales is important for understanding the dynamics involved. In response, we have added a sentence to clarify the seasonal timescale specifically. Rather than removing the original sentence, we felt a brief explanation would better address the concern and enhance the clarity of the introduction.

Changed to: "Additional seasonal processes modulating AT and CT include brine rejection and sinking of  $CO_2$ -rich brine (Rysgaard et al., 2011), air-ice  $CO_2$  fluxes controlled by sea ice permeability (Geilfus et al., 2012), and bacterial respiration of dissolved organic carbon incorporated into sea ice (Zhou et al., 2016). However, over an annual cycle, sea ice does not serve as a net source or sink of these species to the underlying seawater. Instead, it functions as a mechanism for temporal and spatial redistribution, as these species are trapped during ice formation in autumn and winter, and subsequently released during ice melt, potentially at a different location due to drifting sea ice (Thomas et al., 2011)."